

# (D)rifting in the 21st century: Key processes, natural hazards and geo-resources

Frank Zwaan[1, 2], Tiago Alves[3], Patricia Cadenas[4, 5], Mohamed Gouiza[6], Jordan Phethean[7], Sascha Brune[1, 8], and Anne Glerum[1]

[1]GFZ German Research Centre for Geosciences, Potsdam, Germany
[2]University of Fribourg, Fribourg, Switzerland
[3]3D Seismic Lab, School of Earth and Environmental Sciences, Cardiff University, Cardiff, United Kingdom
[4]Memorial University of Newfoundland, St. John's, Canada
[5]FCiências, University of Lisbon, Lisbon, Portugal
[6]Computational Infrastructure for Geodynamics (CIG), University of California, Davis, United States of America
[7]University of Derby, Derby, United Kingdom
[8]University of Potsdam, Potsdam, Germany
Correspondence: Frank Zwaan (frank.zwaan@gfz-potsdam.de)

**Abstract.** Rifting and continental break-up is a key research topic within geosciences, and a thorough understanding of the processes involved, as well as of the associated natural hazard and natural resources is of great importance to both science and society. As a result, a large body of knowledge is available in the literature, yet most of previous research focuses on tectonic and geodynamic processes and their links to the evolution of rift systems. However, we believe that the key challenge

for researchers is to make our knowledge of rift systems available and applicable to face new societal challenges. In particular, we should embrace a system analysis approach, and aim to apply our knowledge to better understand the links between rift processes, natural hazards, and the geo-resources that are of critical importance to realize the energy transition and a sustainable future. The aim of this paper is therefore to provide a first-order framework for such an approach, by providing an up-to-date summary of rifting processes, hazards, and geo-resources, followed by an assessment of future challenges and opportunities for

research. We address the varied terminology used to characterise rifting in the scientific literature, followed by a description of rifting processes with a focus on the impact of (1) rheology and stain rates, (2) inheritance in three dimensions, (3) magmatism, and (4) surface processes. Subsequently, we address the considerable natural hazards and risks that occur in rift settings, which are linked to (I) seismicity, (II) magmatism, and (III) mass wasting, and provide some insights in how the impacts of these hazards can be mitigated. Moreover, we classify and describe the geo-resources occurring in rift environments as (a) non-energy

resources, (b) geo-energy resources, (c) water and soils, and (d) opportunities for geological storage. Finally, we discuss the key challenges for the future linked to the aforementioned themes, and identify numerous opportunities for follow-up research and knowledge application. In particular, we see great potential in systematic knowledge transfer and collaboration between researchers, industry partners and government bodies, which may be the key to future successes and advancements.





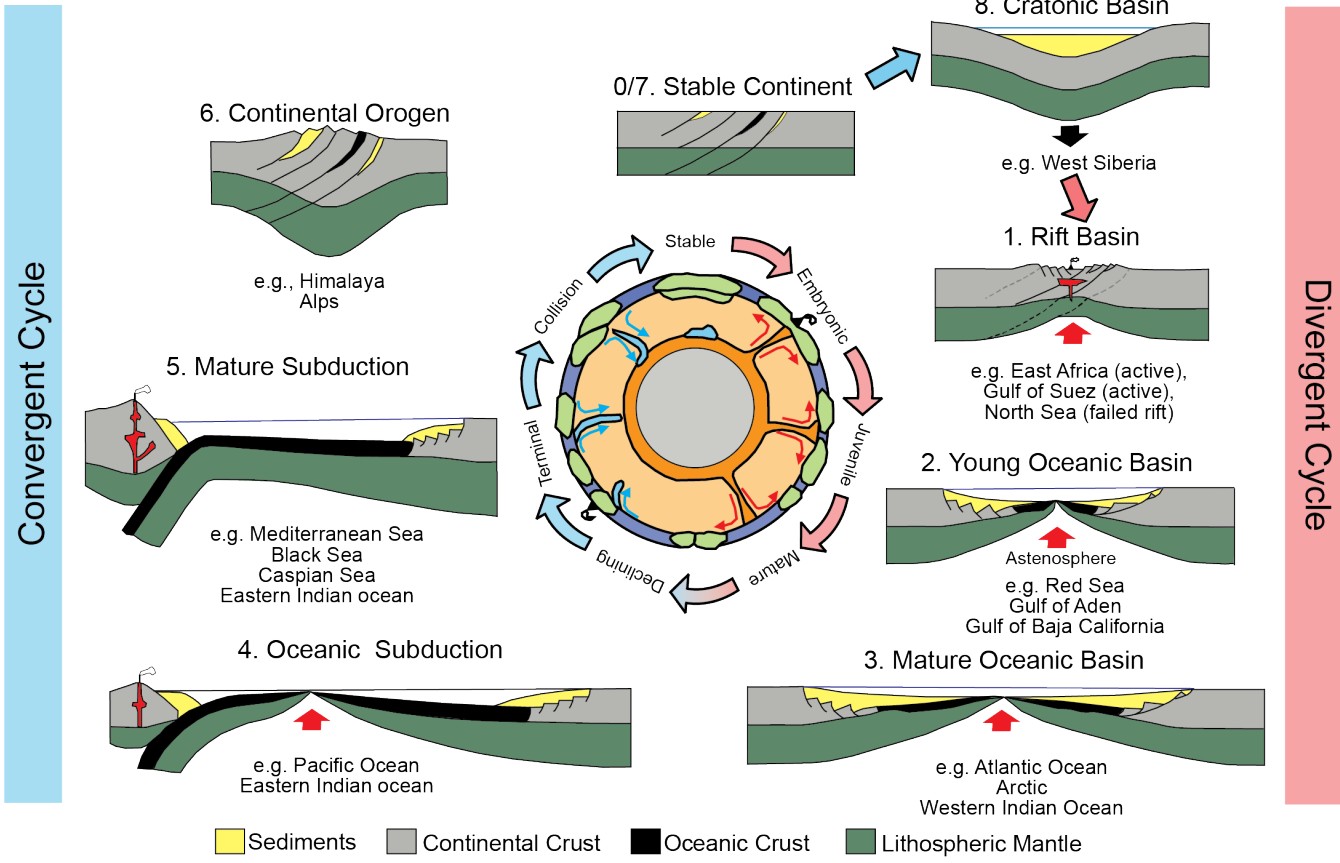

**Figure 1.** Schematic representation of the Wilson cycle, as originally proposed by Wilson (1968). Image modified after Wilson et al. (2019).

# 1 Introduction

The surface of our planet is in perpetual motion as the Earth's continents continuously converge and diverge, assembling supercontinents through subduction and collision, or separating them through rifting and continental break-up processes. This constant assembly and dismantling of continents, known as the Wilson Cycle (e.g., Wilson, 1966, 1968; Wilson et al., 2019) (Fig. 1) is driven by plate tectonic forces that act on the Earth's lithosphere (i.e., the rigid outermost layer of our planet made up of tectonic plates, situated above the asthenospheric mantle, Fig. 1). Subduction and collision of tectonic plates during the 25 convergence phase of the Wilson Cycle have created the world's mountain ranges and deep oceanic trenches. By contrast, rifting and continental break-up during the divergence phase of the Wilson Cycle involve the initial formation of individual rift basins, which can gradually link up to evolve into large-scale rift systems. They may ultimately form oceanic basins when break-up of the continental lithosphere takes place. These oceanic basins are flanked by rifted continental margins on either side and comprise a mid-oceanic spreading ridge in between them, along which new oceanic lithosphere is generated (Fig. 1).

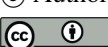

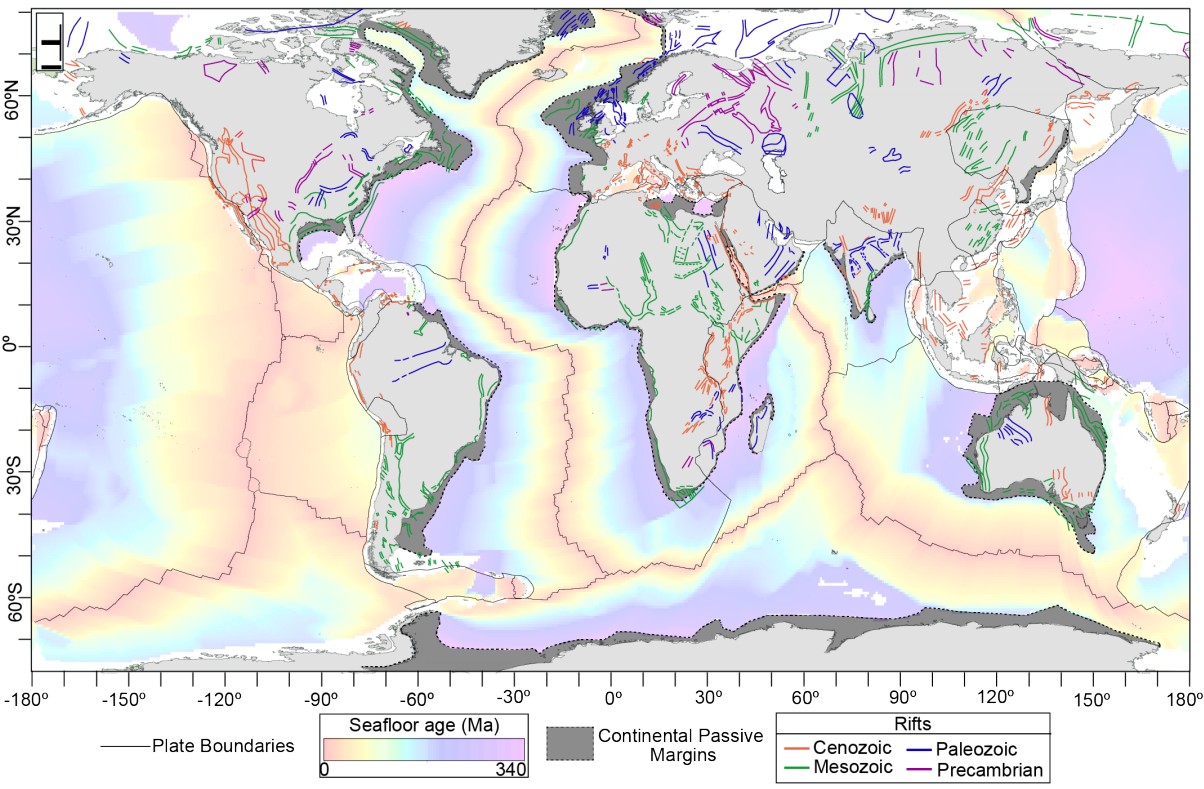

**Figure 2.** Top: Global distribution of continental rift basins and rifted continental margins. Rift distribution after Şengör (2020). Rifted Continental Margins after Berndt et al. (2009). Plate boundaries from Bird (2003) taken from www.earthbyte.org. Seafloor age from Müller et al. (2019) and Seton et al. (2020).

A detailed understanding of the geodynamic processes involved in rifting and continental break-up is of great scientific, economic, and societal relevance. Onshore and offshore rift basins contain extensive sedimentary accumulations that archive crucial information regarding past local and global environmental changes, as well as vast amounts of geo-resources, i.e. natural resources such as hydrocarbons, geothermal energy, helium and natural hydrogen gas ($H_2$), mineral and non-mineral deposits, fresh (ground)water and fertile soils (e.g., Catuneanu et al., 2009; Davison and Underhill, 2013; Zappettini et al., 2017). On the
other hand, rift basins cover large swaths of the Earth's surface (Fig. 2), and present significant hazards in relation to volcanism, earthquakes, and (submarine) landslides, especially since large populations are concentrated in such environments (e.g., Brune, 2016). In fact, natural disasters in rift environments have claimed thousands of lives over the course of human history.



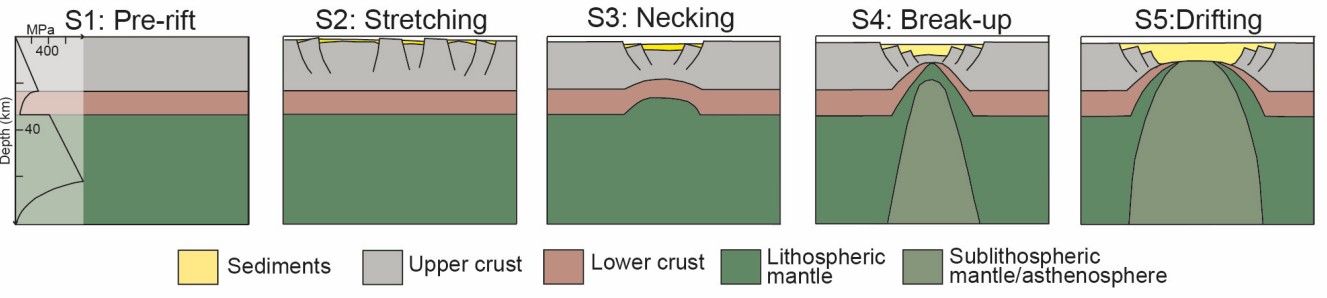

**Figure 3.** Schematic representation of the key rifting stages.

Due to the importance of rifting, an extensive body of scientific literature is available on the topic, summarised and reviewed by numerous authors (e.g., Bradley, 2008; Corti, 2012; Roberts and Bally, 1012; Franke, 2013; Nemčok, 2016; Alves et al., 2020; Şengör, 2020; Buiter et al., 2022; Brune et al., 2023; Peron-Pinvidic, 2022; Pérez-Gussinyé et al., 2023). Most of these authors have focussed on tectonic and geodynamic processes and their links to the evolution of rift systems. However, we believe that the key challenge laying ahead for researchers is to make our knowledge of rift systems available and applicable to new societal challenges. In particular, we should aim to apply our knowledge to better understand the links between rift processes, natural hazards, and the geo-resources that are of critical importance to the global transition towards a sustainable future economy.

The aim of this paper is therefore to provide an up-to-date overview of rifting research with this challenge in mind. We first describe the various definitions used to analyse the evolution of rift systems from inception to break-up and drifting. Subsequently, we focus on the key processes that control the evolution of rift systems, and link these processes to the occurrence and development of natural hazard and geo-resources in such systems. By doing so, we explore key challenges and opportunities for new research efforts and knowledge application, and we hope that this paper as a whole will serve as a guide inspiring upcoming projects in the field of rift research.

## 2 Natural rifting processes

In this section we summarise the key processes causing and influencing the development of rift systems. We start by defining rift terminology applied in the field, before addressing how the lithosphere is deformed as a result of rifting. We subsequently explore the impact of structural inheritance and the need to understand rift systems in a 4D framework (i.e., in time and space). The final topics we treat are the impact of magmatism and surface processes on the evolution of rift systems.

### 2.1 Terminology

The study of rifting has a long history, reaching back to the late 1500's, when cartographers first noticed the apparent fit between the coastlines on both sides of the South Atlantic (Romm, 1994). A broad variety of methods has been applied to the study of



rifts over the past centuries. These methods include geological mapping and sampling, borehole logging, interpretation of 2D
and 3D seismic datasets, aerial and satellite observation, as well as analogue and numerical modelling of rifting processes. As
a consequence of using different analytical and exploratory methods in distinct areas of the world, researchers have historically
developed a plethora of overlapping terminology to describe rifting processes and the different stages of rift evolution, which
ranges from initial thinning of the lithosphere and the associated formation of rift basins, to the eventual development of

rifted margins flanking oceanic basins (e.g., Corti, 2012; Péron-Pinvidic and Manatschal, 2009) (Fig. 3). In this paper, the
term "rifting" refers to the development of an extensional tectonic setting due to divergent tectonic plate motion, be it in an
continental or oceanic environment (i.e. before and after break-up of the continental lithosphere and the establishment of an
oceanic lithosphere, respectively). Similarly, we use the term "rift system", "rift environment" or "rift settings" for extensional
tectonic systems in both continental and oceanic contexts. We also apply the broad term "rifted margin", where synonyms are

"passive margins" (in contrast to active subduction margins, even though "passive" rifted margins are often actively deforming),
"extensional margins", "divergent margins" or "Atlantic margins" (in contrast to the Pacific subduction margins). In essence,
we adopt the general classification used by Péron-Pinvidic and Manatschal (2009), recognising five main stages during the
evolution of rift systems (Fig. 3). These five main stages serve as a robust first-order framework throughout this work:

– The initial state of the system prior to rift initiation is defined as Stage 0 (Pre-rift).

– As soon as rifting is initiated, the system moves into Stage 1 (Stretching), when extension is accommodated by distributed
deformation of the continental lithosphere, leading to widespread normal faulting at the surface.

– During subsequent Stage 2 (Necking), extension starts to heavily localise, causing a strong localised thinning of the
continental lithosphere, the development of a distinct rift basin, and a marked rise of the mantle below.

– The start of Stage 3 (Break-up) is marked by the separation of crustal layers following hyperextension (high degrees of
crustal thinning) of the rifted margins, exhumation of mantle material to (or near to) the surface, and the development of
the Continent-Ocean Transition (COT) when the first oceanic lithosphere is formed.

– Finally, Stage 4 (Drifting) represents rifting after the establishment of a mid-oceanic spreading ridge and the development
of an oceanic lithosphere. Rifted continental margins are fully developed by this stage and generally record regional-
scale subsidence and seaward tilting due to both cooling of the lithosphere and loading induced by the development of
extensive sediment deposits.

It must be noted that we consider these five stages to be representative of the general evolution of rift systems, independent
of the tectonic environment in which they develop; rifting may be caused by for instance mantle plume activity (active rifting),
far-field stresses (passive rifting), subduction rollback (back-arc rifting) or continental collision (orogenic-induced rifting).
These different causes leading to rifting have inspired multiple classifications in the literature (e.g., Merle, 2011; Şengör, 2020;
Peron-Pinvidic, 2022), yet the expression of rifting processes generally follows the same general trend as outlined above.

Even so, lithospheric extension may halt at any stage prior to the drifting phase, resulting in the so-called failed rifts (aulaco-
gens) which, nonetheless, are affected by post-rift thermal sagging. If a subsequent phase of convergence ensues, tectonic



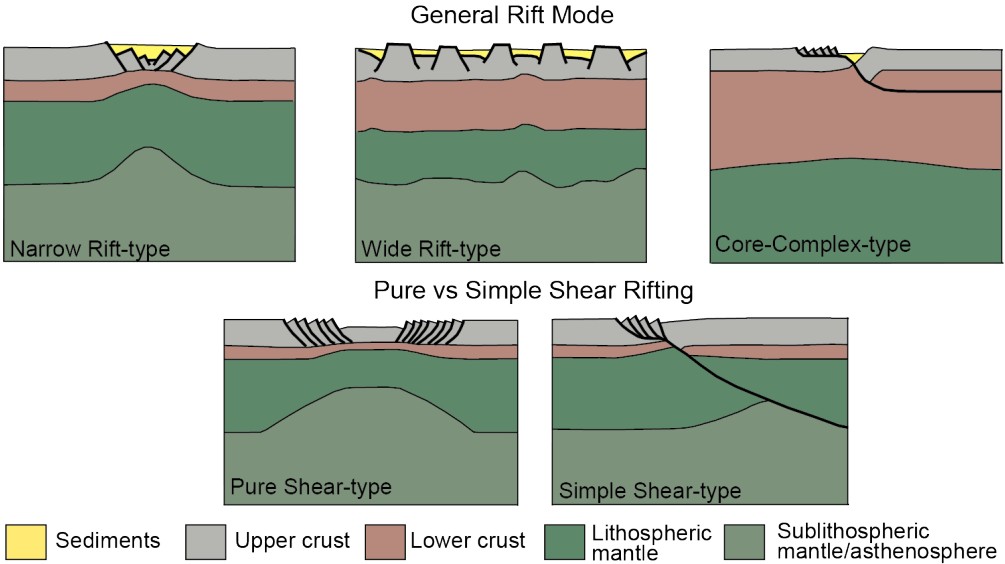

**Figure 4.** Top row : Narrow, wide and core-complex rifting modes (e.g., Buck, 1991; Brun et al., 2018). Bottom row: Pure shear versus simple shear rift geometries, also known as the McKenzie (1978) and Wernicke (1985) models, respectively.

inversion in such failed rifts occurs (e.g., Zwaan et al., 2022, and references therein) and, if this convergence continues in time, a short-cut version of the Wilson Cycle may be established (Chenin et al., 2019). Furthermore, various "passive" rifted margins

are unusually uplifted due mantle processes (e.g South African and Afar-Red Sea margins, Lithgow-Bertelloni and Silver, 1998; Hassan et al., 2020), magmatic underplating (e.g., the Brazilian, Madagascar and Western Indian margins Gunnell and Fleitout, 1998; Mohriak et al., 2008; Radhakrishna et al., 2019), plate convergence (e.g., the Moroccan Atlas and its inverted rift basins, Benabdellouahed et al., 2017), or glacial rebound after the recent removal of thick ice sheets (e.g., the Norwegian margin, Bungum et al., 2010).

Moreover, in the literature, another key distinction has been established between magma-rich and magma-poor rifts systems, since large-scale magmatism, often linked to mantle plume activity has a significant impact on rifting per se, as well as on rifted margin architecture (e.g., Franke, 2013; Buck, 2017; Sapin et al., 2021). This distinction is further explored in section 2.4.

## 2.2  Impact of rheology and strain rate

The type of lithospheric deformation occurring in a rift system will largely control its evolution. The general type of deformation is largely dependent on the interplay between lithospheric rheology, timing (the specific rifting stage), and plate motion velocities. Within this context, rifting is characterised by thinning of the lithosphere, accommodated by either ductile flow or brittle normal faulting.





Firstly, the overall rheology of the lithosphere is strongly impacted by the presence of weak ductile layers in the lithosphere,
most significantly of which is the ductile lower crust (e.g., Brun, 1999; Burov et al., 2006, Fig. 4), though clay and evaporite
layers can have a similar effect on a smaller scale (section 2.4). When present, as in standard continental lithosphere with some
40 km of crust on top of 100 km mantle lithosphere, such a layer can decouple brittle deformation in the mantle lithosphere,
which represents the strongest (i.e., most competent) part of the lithosphere, from brittle deformation in the overlying, com-
petent upper crust. Increased decoupling in hotter lithosphere means that deformation is free to localise throughout the upper
crust, and parts of the upper lithospheric mantle, leading to distributed or wide rifting, and even core complex development
(e.g., for instance in the Basin and Range province in the USA, or the Aegean Sea; Kydonakis et al., 2015; Brun et al., 2018,
Fig. 4). By contrast, systems with a thin or no weak ductile lower crust, such as cold cratonic lithosphere, record most (if not
all) of a lithospheric column that is strong and brittle, rendering structural decoupling a minor factor. As a result, the deforma-
tion type is fully dictated by the mantle lithosphere, and tends to be localised (narrow rifting, e.g., Brun, 1999; Brune, 2016),
or simply occurs wherever the lithosphere is less strong, as documented in the East African Rift System (see section 2.3).

Secondly, the progression of rifting is of importance as the rheology of the lithosphere changes over time. A very thick
ductile lower crust that can lead to core complex formation is characteristic of systems with thickened crust, for instance
after the development of a mountain range in which a surplus of radiogenic heating occurs prior to rifting, i.e. during the
pre-rift stage (Fig. 3) (Buck, 1991; Brun et al., 2018). A typical continental lithosphere containing a weak ductile lower crust
promotes a well-distributed deformation style, typical of the first rifting stage (Stretching stage, Fig. 3). As rifting progresses
the lithospheric layers, including the lower crust, start to thin, so the decoupling effect decreases in importance and the mantle
influence on upper crustal deformation starts to increase, leading to a strongly localised stretching regime, heralding the start of
the second rifting stage (Necking stage, Fig. 3). During this stage, we also tend to observe a shift from initial symmetric, pure
shear rift geometries to asymmetric simple-shear rift geometry (Lavier and Manatschal, 2006) (Fig. 4), although magmatism
can avoid this shift from occurring (see section 2.4). As rifting progresses, the crustal layers will be broken apart to give way
to the rising mantle, so that the influence of the weak ductile lower crust will be negligible from this point in time (Break-up
stage, Fig. 3). Deformation in an oceanic rift system (Drifting stage, Fig. 3) is then mostly controlled by the rheology of the
oceanic lithosphere.

The influence of strain rate is expressed in the altered behaviour of ductile layers in the lithosphere, given that the brittle
deformation of competent lithospheric layers is not strain-rate dependent. When strain rates are low, ductile materials in the
lithosphere generally weaken, whereas higher strain rates cause them to become more competent (e.g., Brun, 1999). This has
a direct impact on the decoupling caused by a weak ductile lower crust during the Stretching and Necking stages: slower plate
motion tends to reduce coupling, promoting wide rifting, whereas faster plate motion increases coupling, promoting narrow
rifting (Fig. 4). During the Drifting stage, slow plate divergence weakens the ductile lower part of the lithosphere in a similar
way as a thick weak ductile lower crust would do, allowing for oceanic core complex formation (Brun et al., 2018). Examples
are found in the Indian SW Indian Ridge as well as the Gakkel Ridge in the Arctic Ocean (Dick et al., 2003; Zhou et al., 2022).
By contrast, fast plate motion during Drifting leads to more typical mid-oceanic ridges with well-defined axial valleys through
strengthening of the lithosphere, forcing a narrow rifting style, for instance along the Mid-Atlantic Ridge (Macdonald, 2019).



It should also be noted that plate motion rates in various rifts (e.g., the Atlantic rift, the Australia-Antarctica rift and the South
China Sea) tend to strongly increase some 10-20 Myr prior to continental break-up (Brune, 2016). This increase in plate motion
velocity from ca. 5 mm/yr (e.g., in the present-day continental East-African Rift System, Stamps et al., 2021), up to some 20
mm/yr or more (e.g., in the Red Sea and Gulf of Aden oceanic basins, ArRajehi et al., 2010), is linked to strong weakening of
the overall lithosphere during the preceding Necking stage. Consequently, any resistance to the plate tectonic forces acting on
the rift system is largely removed, allowing for the plate motion to accelerate so that the plates can attain the "real unrestrained"
plate motion velocity typical of rift basins in the Drifting stage (Brune, 2016).

Furthermore, the spherical shape of the Earth dictates that the motion of tectonic plates involves rotation about an Euler
pole. Such plate rotation induces significant gradients in plate divergence velocities along plate boundaries, and thus along rift
systems (e.g., ArRajehi et al., 2010; Stamps et al., 2021), with various impacts on the development of rift segments. Firstly,
these plate divergence gradients induce rift basin formation far away from the Euler pole, followed by rift basin propagation
towards the same Euler pole, as observed along the East African Rift (e.g., Biggs et al., 2021; Zwaan and Schreurs, 2023).
Secondly, such plate divergence gradients cause along-strike differences in tectonic style. For instance, the aforementioned
ultraslow Gakkel Ridge with its core complexes is close to the North America-Eurasia Euler pole, whereas further away from
the pole divergence velocities increase and we find the well-defined axial valleys of the Mid-Atlantic Ridge. The Gakkel Ridge
also shows an oceanic rift system propagating into continental lithosphere, at the Laptev Margin, where the relatively well-
localised deformation of the Gakkel Ridge is dispersed over various small basins (Franke et al., 2001). Similar settings are
found in the Havre Trough/Taupo Rift system in New Zealand (Benes and Scott, 1996), and in the Woodlark Basin in offshore
Papua New Guinea (Taylor et al., 1999).

## 2.3 Inheritance and rifting in 3D

The long and complex history of the earth's continental lithosphere leaves us with various types of inheritance that weaken the
lithosphere and may affect strain localization during rifting. Structural inheritance can come in the shape of pre-rift structures
such as discrete faults or shear zones, pervasive fabrics in basement rocks, variations in lithospheric strength or layering
between for instance a craton and adjacent terranes, compositional variations due to chemical alteration, or thermal variations
due to mantle activity or previous thinning (Schiffer et al., 2020; Glerum et al., 2020; Gouiza and Naliboff, 2021; Samsu et al.,
2022).

The reactivation of inherited lithospheric structures during rifting depends on various factors. Most importantly, the weakness
must sufficiently impact the strength of the lithosphere; inherited faults that are poorly developed are shown to have little
impact on subsequent basin development, and vice versa for well-developed faults, as for example observed in East Africa and
in the Trans-Mexican Volcanic Belt (e.g., Maestrelli et al., 2020; Wang et al., 2021). Even large-scale suture zones may not
always reactivate, as Wilson (1966) recognised in the North Atlantic realm. In such a setting, the location of the weakness in
the lithosphere, and the rheology of the lithosphere is of importance; when a weak ductile lower crust is present, structural
inheritance in the lithospheric mantle and crust may reactivate independently due to structural decoupling (Zwaan et al., 2022).
Conversely, in case of high coupling, inheritance in the competent mantle lithosphere should be expected to have a dominant





control on the localisation of deformation. Increased structural coupling due to progressive rifting can lead to the overprinting of crustal structures by subsequent mantle-controlled deformation during necking (e.g., Zwaan et al., 2022), an example of
which is recorded in the Mesozoic North Sea Rift (Erratt et al., 1999).

Moreover, the 3D orientation of the structural inheritance is of importance; even a well-developed inherited shear zone may not reactivate, when not oriented favourably for reactivation in terms of dip and strike, with respect to the regional stress field (e.g., Maestrelli et al., 2020; Bonini et al., 2023). As a consequence, the structural inheritance the individual rift segments follow (e.g., along the various branches of the East African Rift System, or along the coast of the Atlantic, Philippon and Corti, 2016;
Brune et al., 2016), can result in different structural styles along each rift segment, from orthogonal rifts to oblique and even transform systems (e.g., Corti et al., 2007; Agostini et al., 2011). Where orthogonal rifts display along-strike faulting, oblique rift systems tend to develop oblique (en echelon or offset) fault systems and rift basins (e.g., the Lake Baikal rift zone and the Main Ethiopian Rift, Petit et al., 1996; Agostini et al., 2011). Highly oblique rift systems may develop into transform margins after the break-up stage, as observed along the Knipovich Ridge (Dumais et al., 2020) and the Davis Strait (Hosseinpour et al.,
2013), both in the North Atlantic.

Finally, some types of inheritance may actively prevent reactivation. Rifting is often multi-phased, with potential long periods of tectonic quiescence in between (e.g., Doré et al., 1999). This leaves time for cooling and strengthening of the mantle material that has risen below a rift basin that entered the Necking stage, which may prevent the reactivation of that basin when the next rifting phase starts. Instead, the reactivated rift system can jump to localise along a different rift axis, where the lithosphere is
weaker, as highlighted by the sequence of pre-break-up rift events with different rift axes in the Mesozoic of the North Atlantic realm (e.g., Doré et al., 1999; Naliboff and Buiter, 2015).

## 2.4 Magma-rich and magma-poor rifting

Multiple rift systems have experienced extensive magmatism at some point during their development, which classifies them as magma-rich systems (in contrast to magma-poor systems, e.g., Franke, 2013; Tugend et al., 2015). Magmatism is often the
result of mantle anomalies (Peace et al., 2020), and the timing of magmatism is of key importance to understand its impact on the evolution of rifting, continental break-up, and rifted margin architecture (e.g., Buiter and Torsvik, 2014; Sapin et al., 2021).

Some rift systems experienced magmatism prior to rifting (i.e. during the pre-rift stage). A prime example is found in the Afar triple junction area, where extensive flood basalts up to 2 km thick covered large areas of present-day Ethiopia, Eritrea and Yemen (Mohr, 1983). This outpouring of flood basalts is linked to the arrival of one or multiple mantle plumes below
East Africa (Hansen et al., 2012; Hassan et al., 2020, and references therein). In general, syn-rift volcanism can occur during all stages of rifting (Peace et al., 2020; Manatschal et al., 2023). A key impact of such syn-rift magmatism is the significant reduction of lithospheric strength that can occur when intruded by magma (Buck, 2006) (Fig. 5). This strength reduction allows for efficient localisation of deformation along the rift axis, accompanied by only limited faulting, altogether referred to as "magma-accommodated rifting", which allows rifting to remain rather symmetrical. Deformation along the rift axes in the
Ethiopian Rift and Afar is considered magma-accommodated, and has a very specific style that can also be found in Iceland (Rime et al., 2023, and references therein). The Afar region also shows the various stages of magma-rich rifting up to ongoing





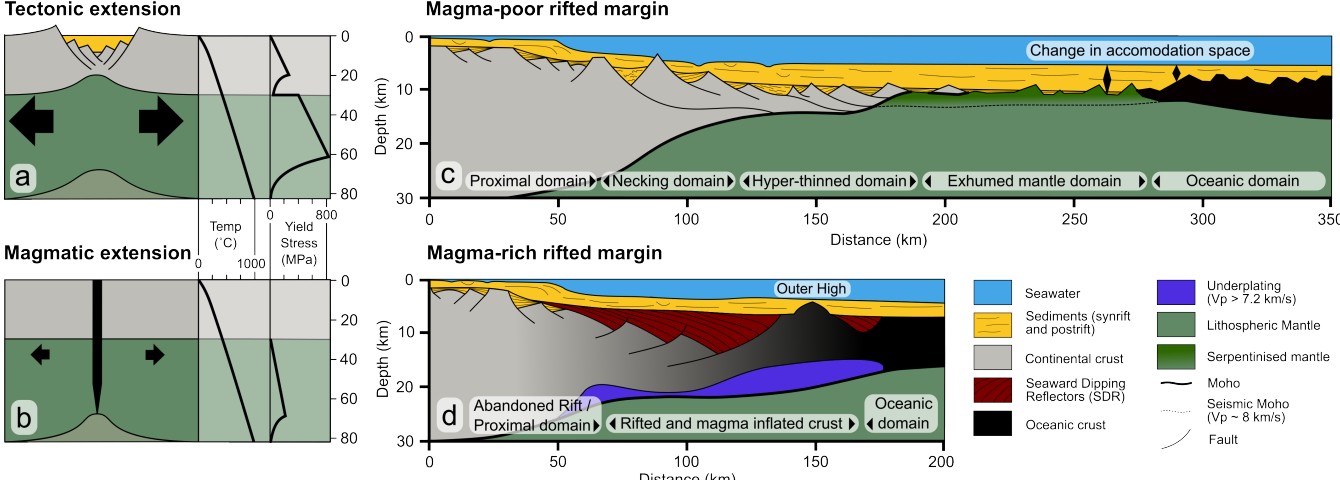

**Figure 5.** Schematics of rifting without (a) and with (b) contemporaneous magmatic intrusions, redrawn from Buck (2006). The yield stress required for extension to progress is significantly lower where magmatic injections weaken the lithosphere. The related characteristics of magma-poor and magma-rich rifted margins, which form during rifting without and with contemporaneous magmatism, are shown in panels (c) and (d), respectively.

break-up along the so-called magmatic segments that mark the rift axes (Varet, 2018). On the other hand, (mafic) intrusions, when allowed to cool and solidify, can strengthen the lithosphere (Liu and Furlong, 1994), somewhat similar to the cooling of shallow mantle material below failed rifts (see section 2.3).

As magma-rich systems reach their break-up configuration (Stage 3), they typically develop large-scale lava flow complexes that dip seaward, which stand out on seismic lines (hence the term seaward-dipping reflector or SDR), accompanied by continent-ward dipping normal faults (Franke, 2013) (Fig. 5). The loading and flexure induced by these lava flows and intrusions may cause the general curved seaward-dipping nature of the SDRs, as well as the continent-ward dipping normal faults observed in such systems (Wolfenden et al., 2005; Buck, 2017).

## 2.5    Erosion and sedimentation

As rifts evolve, erosion and sedimentation (surface processes) actively shape their surface features. The associated transport and redistribution of lithospheric material can in turn influence large-scale tectonic processes, by sedimentary loading in the basins and unloading of the eroding highs (Fig. 6, Burov and Cloetingh, 1997). These can also modify the thermal profile of the lithosphere, and the deposition of evaporites that enable salt tectonics.

Erosion of rift shoulders and other highs in a rift system removes the shallower units and exposes underlying rock units, especially when deep river incisions occur as those of the Blue Nile on the Ethiopian Plateau (Ismail and Abdelsalam, 2012). Sedimentary loading in rift basins induces enhanced subsidence of the graben valley floor, causing large normal faults to remain active for longer (Olive et al., 2014; Neuharth et al., 2022), and can induce ductile flow in the lower crust in relatively hot rift




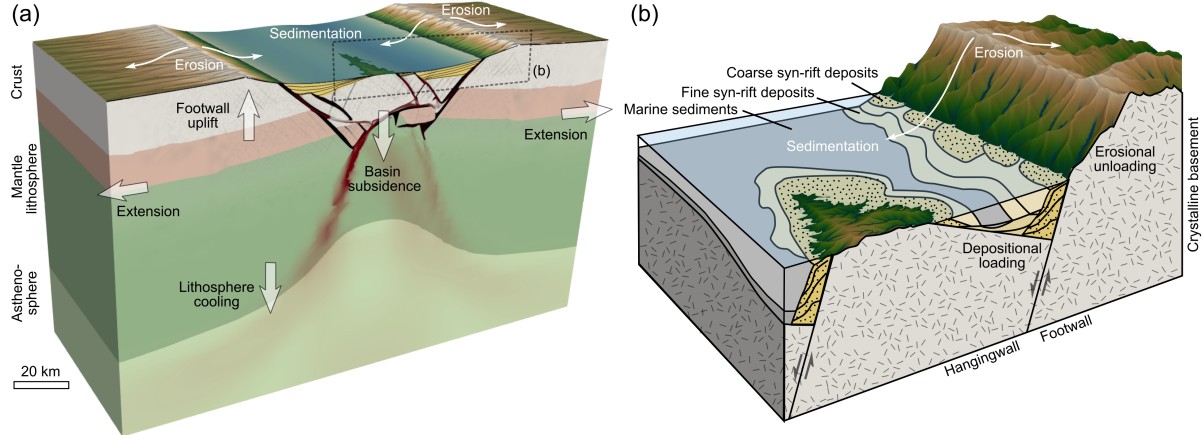

**Figure 6.** Surface processes and tectonic deformation. (a) Lithospheric thinning leads to isostatic subsidence of the surface, which creates accommodation space for sediments. At the same time, flexural uplift of rift shoulders generates steep erodible slopes that provide a source of clastic sediments. Image based on coupled numerical models of geodynamics and landscape evolution (Neuharth et al., 2022). (b) Conceptual model of a deep syn-rift basin. Surface processes feedback on tectonic deformation by unloading the footwall via erosion while simultaneously loading the hanging-wall through sedimentation. Sediment grain size generally decreases with distance from the coast, but is highly susceptible to changes in sea level, tectonics and river network dynamics. Grain size exerts first-order control on rock permeability and fluid circulation, a key process for the formation of geo-resources (Section 4). Image inspired by Gawthorpe and Leeder (2000).

settings (e.g., Clift et al., 2015). The stage of rifting, location in the rift basin and the provenance of the sediments largely

determine the sedimentary infill of a given site, which, nevertheless, follows a large-scale transgressional sequence as a rift system evolves. Stretching (Stage 1) sedimentation is likely continental or lacustrine, with coarse-grained deposits close to exposed fault scarps. During Necking (stage 2), quick deepening and drowning of isolated rift basins is likely to occur, with fine-grained and anoxic sediments (including potential hydrocarbon source rocks, see section 4.2.1) being dominant in the basins. Fine-grained deposits continue to dominate the system during break-up (stage 3) and drifting (stage 4). Typically, a

break-up unconformity forms, marking the rearrangement of the system during stage 3 (Chenin et al., 2015; Morley, 2016).

The large-scale deposition of evaporite bodies in rift systems generally occurs during Necking or Break-up, where the former tends to result in restricted evaporite basins, whereas the latter more likely generates large and continuous evaporite deposits (e.g., along the South Atlantic margins, the Gulf of Mexico, and the Red Sea, Rowan, 2014; Augustin et al., 2014). Yet, some large-scale salt deposits were formed in Pre-rift (stage 0) times, e.g., the Zechstein deposits in NW Europe which were

present prior to Mesozoic rifting (Littke et al., 2008), or during the Stretching stage, e.g., along the Iberian and Newfoundland margins (Rowan, 2014). Due to the relative weakness of these evaporite deposits, it can be easily deformed in a ductile fashion, leading to salt tectonics (Fig. 7). Such salt tectonic deformation along rifted margins is driven by margin tilt due to thermal sag, by sedimentary loading, or a combination of both, leading to highly complex structures (Peel, 2014). In systems with pre-rift evaporite basin development, subsequent salt tectonic deformation is driven by both normal faulting in the basement



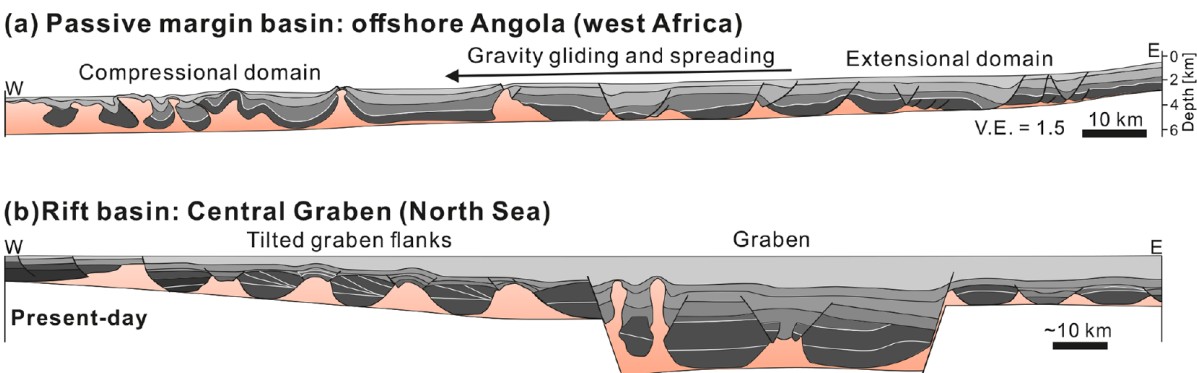

**Figure 7.** Examples of salt tectonics from the Angolan continental margin, and the North Sea Central Graben. The salt (evaporite) layer is indicated in pink. Adopted from Warsitzka et al. (2021).

and basement tilt (Warsitzka et al., 2021). It may be noted that the presence of relatively weak shale deposits can also lead to similar deformation styles as found in salt tectonic settings (e.g., in the Niger delta, Wiener et al., 2011).

Furthermore, the sedimentary infill of rift basins can have a blanketing effect when this infill has a low heat conductivity, leading to a general increase of temperature in the system (Freymark et al., 2017). In turn, this temperature increase on its turn can affect the rheology of the lithospheric layers, potentially increasing decoupling by lowering the strength of ductile layers (e.g., Andrés-Martínez et al., 2019). Sedimentation can also have the opposite effect, as the influx of large amounts of relatively cold material (Wangen, 1995), or material with a high heat conductivity (e.g., evaporites, Duffy et al., 2023). The general influx of sedimentary infill partially can also restore the integrity of the crustal layers, and may even delay continental break-up, as seen in the Gulf of California (Bialas and Buck, 2009).

## 3 Natural Hazards

The various stages of rifting are related to a number of natural hazards that are related to seismicity, volcanism, and mass wasting processes. In this section we describe the different types of natural hazards that occur in rift systems, their origin, frequency and scale, as well as the risk they pose (i.e. their potential impact and human society) and present-day mitigation options.

### 3.1 Seismicity

Wherever active tectonic deformation is occurring, earthquakes are likely to occur as well (Fig. 8). As such, seismicity as the result of sudden displacement along faults in the brittle parts of the lithosphere is to be expected during the whole evolution of rift systems. Although natural seismicity is an integral part of rifting and continental break-up processes, the magnitude of rift-related earthquakes remains often limited to ca. 7 Mw (e.g., Yang and Chen, 2010), in contrast to the mega-earthquakes of Mw >9 recorded in subduction zones (Fig. 8). This discrepancy exists because large earthquakes occur where there is a long



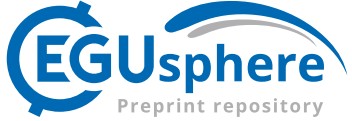

**Figure 8.** Distribution of seismicity on global and regional scale. Crustal normal fault earthquakes with magnitudes Mw≥6.5 are shown as white circles and have been extracted from the GCMT catalogue (time range 1976-2019 https://www.globalcmt.org/CMTsearch.html), based on Neely and Stein (2021). Additional major rift-related earthquakes mentioned in the text are shown as red circles. All normal, thrust, and strike-slip earthquakes with Mw≥6 are depicted as orange circles and are derived from the USGS-ANSS catalogue (time range 1960-2023 https://earthquake.usgs.gov/earthquakes/). Images have been created with GeoMapApp version 3.6.14 (www.geomapapp.org) and include topography data of Ryan et al. (2009). Fault traces are shown as black lines and are based on the GEM Global Active Faults Database (Styron and Pagani, 2020).





and deep fault plane along which the rupture can propagate undisturbed, such as along a subduction interface (Thingbaijam et al., 2017). However, in rift systems, fault planes do not reach as deep as in subduction zones because of a shallower brittle-ductile transition caused by relative hot temperatures at depth. Fault networks in rifts are also less well ordered, because they accumulate significantly less strain when compared to subduction zones. This means there are more disturbances that possibly stop-rupture propagation. Finally, it generally takes less stress build-up to cause displacement in rift systems, relative to compressional systems (Neely and Stein, 2021). As a result, less stress build-up and subsequent stress release, in the form of large earthquakes, is possible in rift systems.

### 3.1.1   Epicentres of rifting-related seismicity

Low extension rates in continental rifts lead to very long recurrence times of large earthquakes, which impedes detailed seismic hazard analysis. It is however clear that rift earthquakes can be devastating as for instance in the East African Rift System (Fig. 8b, Ambraseys and Adams, 1991; Midzi et al., 1999). The 7.4 Ms Kasanga earthquake, one of the largest ever recorded in Africa, rocked the Rukwa Rift area in 1910 (Ambraseys and Adams, 1991), and caused submarine slumps that broke telegraph cables off the coast of Mozambique (see also section 3.3). Casualties were limited though, most likely due to the style of local buildings at the time (Ambraseys and Adams, 1991). Other notable earthquakes with M>6 in the last century include the the 1928 Subukia earthquake in the Kenya Rift (MS=6.9), the 1966 Tooro earthquake in Uganda (MS=6.1), the 1960 Hawassa Earthquake (MS=6.1) and the 1989 Dobi graben event (MS=6.5), both in Ethiopia (Midzi et al., 1999; Zielke and Strecker, 2009). The Afar triple junction region is particularly earthquake-prone (Gouin, 1979; Goitom et al., 2017). Even so, smaller earthquake magnitudes can still have devastating effects as shown by various events in East Africa (e.g., Yang and Chen, 2010; Craig et al., 2011, and references therein).

Significant continental rift-related seismicity is also found outside Africa. In Western Europe, the year 1356 saw the total destruction of medieval Basel, situated at the southern tip of the Upper Rhine Graben (Meghraoui et al., 2001, Fig. 8c). Smaller earthquakes also occur regularly along the Lower Rhine Graben in Germany and the Benelux (e.g., Camelbeeck and Eck, 1994). The Aegean back-arc system is prone to rift-related quakes as well, from the Corinth rift and the area further east around Athens, to the western coast of Turkey (e.g., Akçar et al., 2012; Kapetanidis et al., 2020). Other examples of active continental rift systems posing hazards are the Gulf of Suez rift (Mohamed and Abd El-Aal, 2018), and the Trans-Mexican Volcanic Belt that contains various major cities, among which Mexico City (Maestrelli et al., 2020, and references therein). The Siberian Baikal rift system is seismically active, with many earthquakes exceeding magnitudes of 6. This is due to the fact that the rift is situated in a cold lithosphere, where the brittle-ductile transition is deep and fault planes have a larger extent. The largest recorded Bailkal Rift earthquake even reached a magnitude of Mw=7.8 (the 1957 Muisk event, Doser, 1991), However, as population densities are low in the area, risks are moderate (Arzhannikova et al., 2023). Conversely, in China, exceptionally strong and deadly rift-related earthquakes are known to occur in the Shanxi and Weihe rifts, south-west of Beijing (Xu et al., 2018). These rifts, which have been active since the Pliocene, were the sites of the 1303 Hongdong earthquake and the 1556 Huaxian earthquake, with magnitudes of 8 Mw or higher, and caused over 470,000 and 830,000 fatalities, respectively (Liu et al., 2007).





Furthermore, numerous earthquakes are recorded at mid-oceanic ridges (Fig. 8), yet these pose only limited hazards due to
their distance to population centres. There are situations where oceanic ridges are close to continents, or enter into continental
lithosphere, and where significant earthquakes occur, such as in the Gulf of California (Castro et al., 2021) orthe northern tip
of the Red Sea (Hosny et al., 2013). A special case is found in southern Iceland, where the emerging Mid-Atlantic Ridge runs
close to the area around the capital of Reykjavik, and where historic earthquakes, such as the Ms=6.0 event in 1706, have
caused death and destruction (e.g., Frímann, 2011).

### 3.1.2 Intra-plate earthquakes

The above examples concern actively deforming rift systems. However, rift-related intraplate earthquakes can also occur with-
out active plate tectonic motion, i.e. in stable continental regions away from active plate boundaries that have not seen sig-
nificant deformation over the recent geological past (Schulte and Mooney, 2005). A notorious example is the New Madrid
Seismic Zone in the central USA, which represents a concentration of seismicity linked to the tectonically inactive Palaeozoic
Reelfoot rift and was the locus of the devastating 1811-1812 earthquakes with magnitudes over 7 Mw (Calais et al., 2010).
Other examples include the 1663 Charlevoix earthquake along the St Lawrence rift zone in Quebec, and the 1845 and 1888
earthquakes in the Río de la Plata region, Argentina, which are linked to the inactive Mesozoic Quilmes Trough(Rossello et al.,
2020). Given that large swathes of the earth's continental lithosphere are under some level of stress (World Stress Map, Heid-
bach et al., 2018), and that inactive rift systems may act as inherited weakness zones and are widely distributed (Şengör and
Natal'in, 2001, Fig. 2), large earthquakes may occur at unexpected locations (e.g., Liu et al., 2007).

Furthermore, intraplate earthquakes can strike along rifted margins, which are often somewhat misleadingly referred to as
"passive margins" (see section 2.1). Ongoing deformation caused by various processes, such as magmatic underplating, plate
convergence, or glacial rebound can trigger the reactivation of former rift faults to generate significant earthquake activity. A
notorious example is the magnitude 7.2 Grand Banks earthquake offshore Newfoundland in 1927 (Fine et al., 2005), which
caused a large submarine landslide and an associated tsunami (see section 3.3).

### 3.1.3 Seismic hazard

Research and monitoring allow us to assess hazards, vulnerability and risk of specific rift systems. Seismic surveys and geo-
physical analyses, satellite imagery-based InSAR studies, and measurements of the state of stress in boreholes provide insights
into the local geological setting and the dynamic deformation causing earthquakes (Illsley-Kemp et al., 2018; La Rosa et al.,
2019; Heidbach et al., 2018). Important constraints on earthquake recurrence time can be deduced from historical literature
(e.g., Gouin, 1979), dating of known earthquake-prone faults by investigating sediment deposits adjacent to faults and pa-
leoseismic trenching (Papanastassiou et al., 2005, and references therein), directly dating the fault surface with cosmogenic
nuclide analysis (e.g., Akçar et al., 2012), or by archeological research (Ambraseys, 2006; Aydan and Kumsar, 2015). Rel-
evant data is collected and made available by various organizations, such as the Malawi Seismogenic Source Model in the
EARS (Williams et al., 2022), the EU-funded EPOS Seismology Consortium, the European Facilities for Earthquake Hazard
and Risk (EFEHR) Consortium, or on a global scale by the Global Earthquake Model (GEM) project. These datasets are cru-





cial for disaster management planning, and for the safe development of specific industries, especially those involving nuclear power production and fluid injection (geothermal energy projects and unconventional hydrocarbon production). For instance, the geothermal fluid injection tests in Basel, at the southern tip of the Upper Rhine Graben, caused slip along pre-stressed faults

and minor earthquake damage in the city centre (Mukuhira et al., 2008). Moreover, extensive unconventional hydrocarbon production from sedimentary deposits in a variety of inactive (rift) basins in the USA causes regular seismicity with magnitudes over 5.5 Mw (e.g., van der Elst et al., 2013; Chen et al., 2017).

### 3.2 Magmatism

There is a clear link between rift development and the occurrence of magmatic activity, and even in so-called magma-poor rift

systems, magmatism is present to some extent (see also section 2.4). Magmatism in rifts dominantly occurs due to decompression melting, as the fast upwelling mantle rocks cause temperatures to cross the peridotite solidus, or the solidus of the overlying crustal rocks. This rise of hot material below a rift system can be achieved by either having a mantle anomaly actively pushing upward or by rapid thinning of the lithosphere. The bulk of magmatic material in rift settings remains trapped underneath the crust, in the form of underplating and/or magmatic intrusions. As such, the main hazard posed by rift-related magmatism is

linked to the magmatic material that reaches the surface as volcanic eruptions (i.e., causing lava flows, devastating pyroclastic flows, ejecta, lahars/mudflows, earthquakes, and even tsunamis).

### 3.2.1 Hotspots of rift-related volcanism

Volcanic hazards represents the most risk in rift systems prior to the break-up stage, when the system evolves mostly in subaerial conditions, and where the fertile volcanic soils provide excellent farmland to sustain large human populations. The

most prominent examples of such rift-related volcanism may be found along the highly magmatic Eastern Branch of the East African Rift, where among others Mount Kilimanjaro is situated (Martin-Jones et al., 2020) (Fig. 9). In this context, the Afar Rift at the northernmost end of the Eastern Branch is a special case, as some authors characterise it as an embryonic (subaerial) mid-oceanic ridge (Rime et al., 2023). This highly volcanically active region is mostly a desert at present, with parts situated down to 160 m below sea level (Rime et al., 2023), but contains considerable populations that live with the risks of frequent

volcanic eruptions. For instance, although the 2008 Alu Dalafilla eruption did not cause any human or economic losses due to the remote location of the volcano (Pagli et al., 2012), the 2011 eruption of the Nabro volcano caused several fatalities (Goitom et al., 2015). Situated just in Eritrea the Nabro eruption resulted in the expulsion of 1.5 megatons of $SO_2$, ranking it as the largest eruption since that of Mount Pinatubo in 1991 (Bourassa et al., 2012), and disrupted air travel in the region (Sawamura et al., 2012). The Western Branch also has a moderate amount of magmatism, specifically along the linkage zones between

individual rift segments that allow for easier fluid migration (e.g., Corti et al., 2004, Fig. 9). As large populations are situated along the shores of the various big lakes of the Western Branch of the EARS, the risk posed by volcanism is significant (e.g., Biggs et al., 2021; Hearn, 2022). A Recent example is the 2002 devastation of the city of Goma in the Democratic Republic of Congo by a volcanic eruption of nearby Mount Nyiragongo (Pouclet and Bram, 2021).





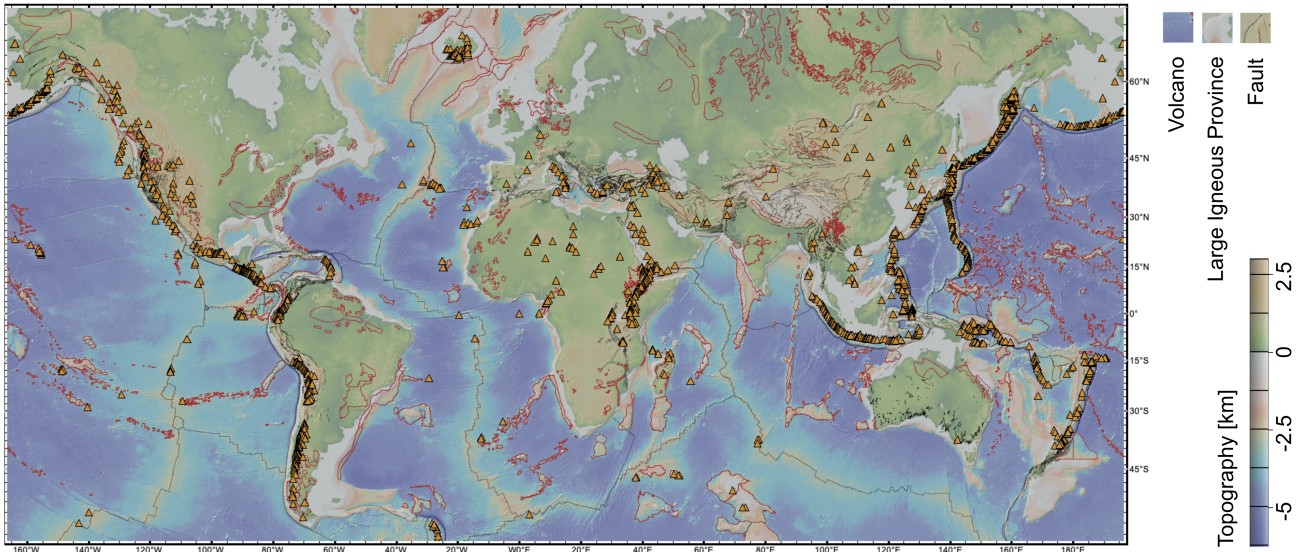

**Figure 9.** Volcanism around the globe, based on a global catalogue of predominantly onshore volcanoes of the Global Volcanism Program (Venzke, 2023), large igneous provinces (Johansson et al., 2018), as well as faults and plate boundaries (Styron and Pagani, 2020). Image has been created via GeoMapApp version 3.6.14 (www.geomapapp.org) based on topography data of Ryan et al. (2009).

The Aegean back-arc system is another location that is prone to continental volcanism, with the 1600 BCE Thera eruption
on the modern-day Greek island of Santorini being perhaps the most notorious volcanic event in the recent geological past of the Mediterranean. This especially violent eruption, one of the largest in recent human history, devastated the island and sent tsunamis throughout the eastern Mediterranean (Karstens et al., 2023), and is thought to have been instrumental in the collapse of the previously flourishing Minoan civilization on nearby Crete (Novikova et al., 2011, and references therein). The Trans-Mexican Volcanic Belt and its numerous volcanic eruptions also pose serious hazards to the various major cities in the
area, including Mexico City (Maestrelli et al., 2020, and references therein). The Taupo Rift Zone, where the Havre Trough back-arc rift system enters the continental lithosphere of New Zealand, is another site of ongoing volcanism that witnessed a supereruption as recent as 26.5 ka (Wilson et al., 2006). Other, volcanic zones in continental rift settings are the Eifel, Massif Central, and Eger Graben areas of the European Cenozoic Rift System, which are less known due to the absence of present-day eruptive activity (Fig. 9). However, the last eruption in the Massif Central only around 8 ka (Merle et al., 2023), whereas
large-scale volcanism in the Eifel area at the northern tip of the Upper Rhine Graben lasted until some 10 ka (Dahm et al., 2020). Also, the Eger Graben farther to the east exhibits geologically recent volcanic activity (Hrubcová et al., 2017). A caveat is that this rift-related volcanism in Western Europe was possibly caused by mantle activity induced by the Alpine orogeny, the tectonic event that is thought to have induced the rifting in Western Europe in the first place (Dèzes et al., 2004; Merle et al., 2023). Even so, there is potential for future volcanism in these areas (Boudoire et al., 2023).





After the establishment of continental break-up, oceanic rift systems tend to be submerged, which reduces the occurrence of volcanic hazards and their impact. Only when a submerged spreading axis is close to the continent, as in the case of the Gulf of California, there could be some risk of an eruption affecting human populations. However, in specific locations, spreading ridges (or spreading-ridge related volcanoes) are subareal, and inhabited. Examples are the volcanic Azores islands on the triple junction between the North American, European and African plates, and most famously Iceland, situated on the Mid-Atlantic

ridge (Gudmundsson, 2007), which is the site of the 2010 Eyjafjallajökull eruption that caused massive disruptions to global air travel (Gudmundsson et al., 2012; Kelman et al., 2023). Next to typical hazards experienced by inhabitants of volcanic islands such as the Azores, the population of Iceland also has to contend with eruptive melting of glaciers sitting on top of their island's volcanoes causing truly massive flash floods that devastate anything in their path (so-called Jökulhlaups; Pagneux et al., 2015). Interestingly, the growth and decline of glaciers on Iceland is linked to the intensity of volcanic activity on the island, as the

associated increase and decrease in glacial load reduces or enhances decompression melting (Cooper et al., 2020). A similar correlation between glacial extent and volcanism has also been noted in Western Europe (Nowell et al., 2006).

### 3.2.2   Volcanic degassing

In addition to the direct volcanism-related hazards such as explosive eruptions, magma-induced earthquakes, lava flows, pyroclastic flows, and lahars/mudflows that can lay waste to large swatches of land (see also section 3.3), magmatic activity in rift

settings is accompanied by substantial hydrothermal circulation and degassing (e.g., Sawyer et al., 2008). These can lead to the development of mud volcanoes and geothermal fields that can pose serious hazards to local populations (e.g., Vereb et al., 2020). Hydrothermal flow can alter the state of stress in the subsurface, causing the activation of faults (see also section 2.1), and degassing can pose major local threats as well, in particular when it comes to $CO_2$; i.e., being heavier than air and scentless, $CO_2$ can accumulate in depressions and suffocate unaware people and livestock (Cantrell and Young, 2009). Furthermore, $CO_2$

(and other gases) can accumulate in vast volumes in the deeper parts of the water column of lakes along rift systems. These dissolved gases in such meromictic lakes may violently escape when the delicate equilibrium in the water column is disturbed due to an earthquake, a landslide, some (minor) eruptive activity, or possibly even without external trigger at all (Schmid et al., 2002; Tassi and Rouwet, 2014). The $CO_2$ released during such limnic eruptions may then spread like an invisible wave through the surroundings, suffocating anything in its path (Gusiakov, 2014). A tragic example occurred at Lake Nyos, one of a series

of volcanic lakes situated along the Central African Shear Zone in Cameroon, where over 1700 people perished in 1986 (Gusiakov, 2014). Similar dangers may be lurking in the great lakes of the East African Rift System, especially since these lakes are much larger than those in Cameroon. A lake of particular concern is Lake Kivu, with millions of people living in its vicinity (e.g., Jones, 2021).

### 3.2.3   Mitigating volcanic hazards

Magmatism in rifts thus poses severe risks, but these can be mitigated to some degree. Induced degassing of meromictic lakes such as Lake Nyos and Lake Kivu reduces the risk of limnic eruptions, and even allows the production of $CH_4$ used for local energy uses (Jones, 2021; Wenz, 2020), (see section 4.2). While degassing or exploring for resources in such lakes, it is crucial





to monitor the equilibrium in the water column so as to not accidentally cause a limnic eruption. Furthermore, there are seismic networks actively monitor earthquakes that could kick off a limnic eruption in meromictic lakes (Oth et al., 2013). We thus have

some means to actively mitigate the risk of limnic eruptions, but preventing volcanic eruptions is an unfeasible proposition, so at present we can only prepare and react when they happen. Similar to meromictic lakes, monitoring of seismic activity, variations in degassing, or changes in ground elevation that may hint at disturbances in the magma chamber and the possibility of an imminent volcanic eruption are key monitoring methods applied in various volcanic rift settings (Biggs et al., 2021; Boudoire et al., 2023). A big challenge may be the recurrence time of eruptions in rifts, that may run in the hundreds of years

(e.g., Nowell et al., 2006; Einarsson et al., 2020). However, in some cases, it is possible to mitigate the potential destruction caused by lava flows by diverting their paths. An intriguing example occurred on the Vestmannaeyjar archipelago, just south of Iceland, where a lava flow on the main island of Heimaey in 1973 threatened to block the harbour entrance. Fishing being the economic lifeline of the inhabitants, the lava flow was successfully prevented from advancing too far by spraying its front with large volumes of sea water (Williams and Moore, 1976).

## 425   3.3   Mass wasting

### 3.3.1   Subareal settings

Rifting causes the development of topography and thus of unstable slopes, either subareal or submarine, that can collapse in mass wasting events. In subareal situations, footwall uplift of major normal faults as well as subsequent erosion can form impressive escarpments, such as those along in the East African Rift System and the Red Sea-Gulf of Aden system, which

in some cases represent some 2 km of topography difference between rift basin and rift shoulder (e.g., Fubelli and Dramis, 2015). Steep slopes and escarpments can be destabilised by mechanical weakening or by ground shaking leading to frequent landslides (Broeckx et al., 2018). The most voluminous landslides in the East African Rift are caused by ground shaking due to earthquakes (Jacobs et al., 2017) or by precipitation-induced weakening (Kropáček et al., 2015; Martínek et al., 2021). Particular areas at risk are the escarpments bordering the Ethiopian highlands (Fubelli and Dramis, 2015; Martínek et al.,

2021), as well as large parts of the Western Branch of the East-African Rift (Stanley and Kirschbaum, 2017), for instance along the shores of Lake Kivu (Jacobs et al., 2018; Depicker et al., 2021).

### 3.3.2   Submerged settings

Other, often more impactful mass wasting events occur offshore, at the continental slopes of rifted margins. Here, large volumes of sediments accumulate over time, since the world's largest rivers drain towards rifted margins. Furthermore, rifted margins

in the high latitudes receive major sediment input due to glacial erosion during ice ages. These large volumes of sediments are deposited at the angle of repose, so that slight destabilisation, either to additional sedimentary loading, earthquakes, or sediment weakening, can cause a catastrophic collapse and massive submarine landslides. Various instances of such collapses and consequent landslides have been recorded by the severing of communication cables over the sea floor (e.g., Ambraseys and Adams, 1991; Assier-Rzadkieaicz et al., 2000). Famously, the magnitude 7.2 Grand Banks earthquake offshore Newfoundland



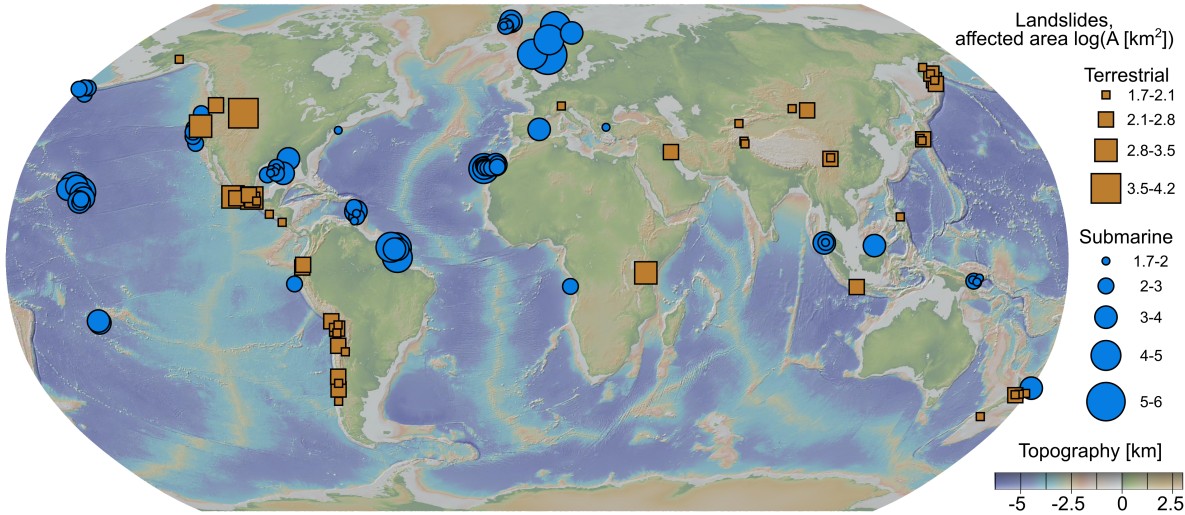

**Figure 10.** World map of giant terrestrial (squares) and submarine (circles) landslides. Rifted margins of the North Atlantic are particularly prone to host large submarine landslides, because they (1) accumulate large volumes of sediments and (2) because they experienced enhanced sediment loading during glacial periods. The size of the symbols codes for landslide surface area. Modified from Korup (2012).

in 1927 caused a massive submarine landslide including turbidite flows that travelled over a distance of over 1000 km at speeds ranging between 60-100 km/h thereby severing telegraph cables (Fine et al., 2005). Communication cables can be easily replaced, but a much more impactful hazard generated by submarine slides are tsunamis, such as the one that accompanied the 1927 Grand Banks earthquake causing 28 victims (Løvholt et al., 2019). Still, perhaps the most well-known submarine landslide is the Storegga landslide that occured along the Norwegian rifted margin in ca. 8,000 BCE (Bondevik et al., 2005).

This tsunami drowned the coasts of Norway, the Shetland and Orkney Islands, Eastern Greenland, and NW Europe down to the Southern North Sea (Nyland et al., 2021). Many other such submarine landslides and associated tsunamis are known from the NE Atlantic (e.g., Leynaud et al., 2009), and from rifted margins around the globe (e.g., Korup, 2012; Thran et al., 2021, Fig. 10). Since many large cities are located along the coast near rifted margins, the impact of tsunamis induced by large submarine slides such as those offshore Norway would be catastrophic. We should however emphasise that the recurrence time of these

huge submarine landslides is very low (Leynaud et al., 2009). Still, global warming and associated potential destabilization of rifted margin increases the risk of landslide tsunamis in the North Atlantic, as well as along other rifted margin settings.

## 4   Geo-resources

Rift systems contain a wealth of geo-resources that will be greatly needed for the energy transition and the establishment of a sustainable economy in the 21st century. In this section we describe (1) the non-energy mineral resources, (2) the various geo-

energy resources, and (3) the temporary and permanent geological subsurface storage options that occur in rift environments.





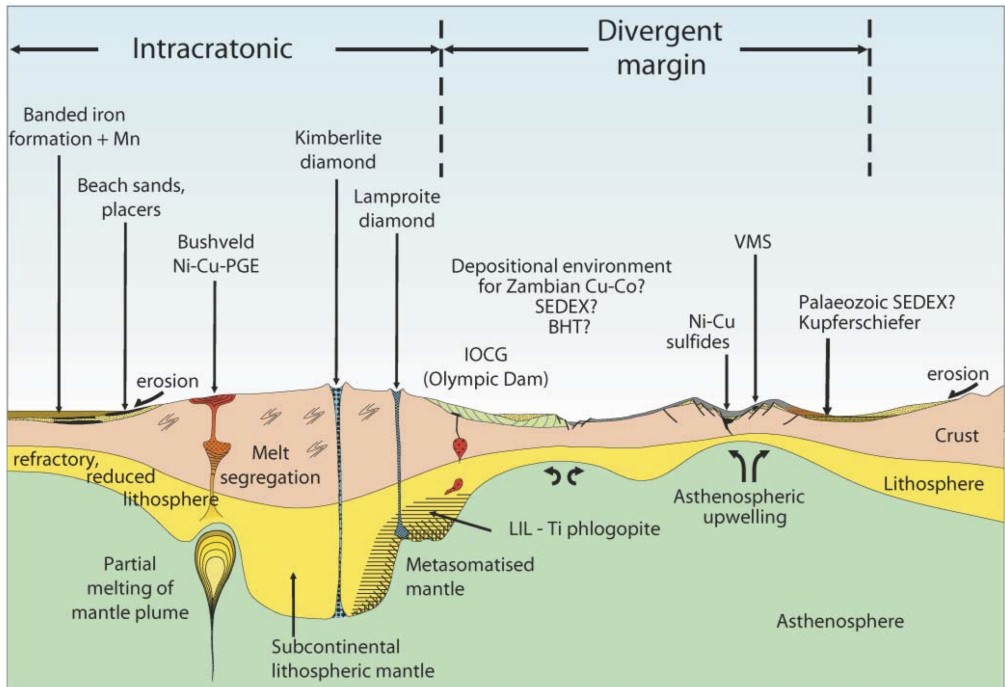

**Figure 11.** Schematic diagram of major mineral deposit types formed in continental crust, and along passive continental margins and oceanic spreading ridges. Adopted from Groves and Bierlein (2007).

## 4.1 Non-energy mineral resources

### 4.1.1 Mineral deposits

Mineral deposits are anomalous concentrations of minerals in rocks or sediments, while ore deposits are those deposits that are economically viable to extract (Heinrich and Candela, 2014). The mineral deposits related to rift systems can be divided into various categories depending on when and how they formed (e.g., Groves and Bierlein, 2007; Zappettini et al., 2017, Fig. 11).

**Pre-rift mineral deposits** are of great importance in rift systems. They come in many shapes and forms, depending on the pre-rift (Stage 0) geological history of the specific setting. The rocks containing pre-rift deposits are exhumed by the combined action of extension tectonics and erosion at the rift shoulders, along individual rift blocks, or along (uplifted) rifted margins. Much of the mineral wealth of the countries along the East African Rift System is based on such deposits exhumed during rifting (e.g., Taylor et al., 2009; Dill, 2007). Other examples of pre-rift deposits are those which are (historically) exploited along the Cenozoic European Rift System (e.g., the Vosges and Black Forest mountains, as well as the Central Massif), where the Hercynian metamorphic basement is exposed (e.g., Forel et al., 2010; Steiner, 2019).

**Sedimentary mineral deposits** are generated by sedimentary processes. One type of sedimentary deposits constitutes deposits in clastic sedimentary environments, called *(paleo)placers*. Placer deposits form through the breakdown of bedrock and





the sorting and concentration of relatively heavy minerals such as diamonds, gold, and platinum-group minerals, by various sedimentary processes (mostly fluvial; Ridley, 2013). Deposits thus form at shorelines and in rivers, and are found for example in the East African Rift System, as well as along the passive margins of East Africa and the Russian Arctic (Dill and Ludwig, 2008; Bochneva et al., 2021). High-grade diamond deposits are found and exploited offshore Namibia (Schneider, 2020). Not all placer deposits are economical, but they can still provide insights into the location of their (pre-rift) origin sites through

provenance analysis (Gong et al., 2021).

*Hydrogene deposits*, chemically precipitated from surface waters, can also be found in rift environments, for example ancient or actively forming evaporites. Evaporites form in compartmentalized basins through solar evaporation and can be mined to produce salt for human consumption or industrial use, as is the practice in present-day Afar and in Lake Magadi in Kenya (Kodikara et al., 2012; Varet, 2018). Depending on the origin of the salt and fluids (marine, hydrothermal, or even from

serpentinization of exhuming mantle material, referred to as "dehydrates", Debure et al., 2019), there may also be important concentrations of potash, lithium, trona and other minerals to be extracted (Kodikara et al., 2012; Bekele and Schmerold, 2020). For instance, phosphorites are phosphorus-rich hydrogene deposits that (bio)chemically form in the shallow marine environment along rifted margins (i.e., on the continental shelf; Ridley, 2013; Kudrass et al., 2017). Given the projected scarcity of phosphorus, a component of fertiliser (Alewell et al., 2020), these relict and recent offshore deposits will be of great

interest to the global community in the near future.

**Sediment-hosted hydrothermal ore deposits** form in sedimentary rift basins both during and after rifting. Hydrothermal fluids circulating through the basins leach metals from deeper in the basin, transport them upwards, and deposit them in sulphide minerals when reaching conditions that trigger metal release (Heinrich and Candela, 2014; Wilkinson, 2014). Temperature, salinity, pH and redox state all affect the solubility of metals in these fluids (e.g., Heinrich and Candela, 2014; Zhong et al.,

2015), making heated saline waters an efficient mineralizing fluid and colder, reduced rocks a possible host (Ridley, 2013; Wilkinson, 2014).

Sediment-hosted mineral deposits are the largest global resource of lead and zinc (Goodfellow et al., 1993; Mudd et al., 2017), base metals important to the energy transition (IEA, 2021). The majority is found in *clastic-dominated (CD) type deposits* (formerly known as SEDEX deposits; Goodfellow et al., 1993) that formed during deposition or diagenesis of the

marine siliciclastic host rock through subseafloor replacement of barite with lead-zinc sulphides (syngenetic to diagenetic mineralization, Leach et al., 2010; Magnall et al., 2020). These mostly Proterozoic and Paleozoic mineralizations occur close to syn-sedimentary faults (e.g., Wilkinson, 2014; Hayward et al., 2021), which could have provided permeable, focussing pathways for upward fluid flow (e.g., Walsh et al., 2018; Sheldon et al., 2019; Rodríguez et al., 2021). Large deposits have been found in the Carpentaria Zinc Belt in Australia, the North American Cordillera, Russia and China (e.g., Leach et al., 2010;

Hoggard et al., 2020). The precursory sediment-hosted barite deposits are also mined in China, India and the US (e.g., Clark et al., 2004).

Contrary to CD-type deposits, *Mississippi-Valley Type (MVT) lead-zinc deposits* formed much later than their host rock (up to tens of millions of years; epigenetic mineralization). The mostly Phanerozoic MVT deposits are found in broad rift basins with thick sequences of shallow marine sediments including platform carbonates that host the mineralization (Ridley, 2013).



These basins have generally been affected by inversion (e.g., Leach et al., 2010), with high topography and/or tectonic loading possibly having driven long-distance fluid flow (Ridley, 2013). By correlating observational datasets, such as of lithosphere thickness, with CD-type and MVT deposit locations, researchers have recently tried to limit the exploration space for these deposits (e.g., Hoggard et al., 2020; Lawley et al., 2022; Burisch et al., 2022).

A big supplier of copper are the *sediment-hosted stratiform copper (and cobalt) deposits* as found in the Kupferschiefer

field in the Zechstein Basin (Poland and Germany), the Central-African Copperbelt, the Kodaro-Udokan basin in Siberia and the White Pine deposit in the USA. These deposits mostly occur at stratigraphic and redox boundaries between terrestrial syn-rift red-bed sandstones (metal source) and overlying shallow-marine or lacustrine organic-rich shales (host rock; Hitzman et al., 2010; Ridley, 2013). Such successions formed in the intracontinental basins of failed rifts, with syn- or post-diagenetic mineralization by oxidised saline fluids happening during the sag or even the inversion phase (Brown, 2014; Hitzman et al.,

520 2010).

Intracontinental basins also contain several types of *sediment-hosted uranium deposits* (e.g., Robb, 2004; Dahlkamp, 2009). For example, epigenetic unconformity-related deposits (such as in the Athabasca Basin, Canada; Jefferson and Delaney, 2007) are found at major unconformities between basements and overlying red-beds (Dahlkamp, 2009). Tabular and roll-front deposits have distinct blanket, respectively, U-shaped geometries and are formed by lower-temperature oxidised fluid circulating

in the sandstones (Ridley, 2013).

An example of **magma-related hydrothermal ore deposits**, volcanic-hosted or *volcanogenic massive sulphide (VHMS or VMS) deposits* formed and currently form through venting of hydrothermal fluids in and at the seafloor (creating for example black and white smokers; Barrie and Hannington, 1997; Hannington et al., 2005). The first such seafloor hydrothermal activity was discovered in the Red Sea oceanic spreading centre (Miller et al., 1966). The majority of active VMS deposits occur along

mid-oceanic ridges, where mostly seawater circulates in the subsurface depositing polymetallic sulphide minerals that contain amongst others copper, zinc, lead, gold and silver (e.g., Shanks et al., 2012; Fuchs et al., 2019). Some fossil VMS systems are found exposed after being obducted by subsequent convergence tectonics. The Oman ophiolite is a well-known example, but the Troodos Ophiolite on Cyprus is world-famous since its VMS deposits have been exploited for thousands of years to the degree that copper (cuprum, in Latin) may have been named after the island, or the other way around (Pirajno et al., 2020).

Examples active VMS systems are common along mid-oceanic ridges, but are also found in the (incipient) spreading centres of Afar (e.g., at Dallol; Varet, 2018).

Diamondiferous kimberlites are a type of **magmatic ore deposit** that contain diamonds. Kimberlites are rocks formed from magmatic ultramafic highly volatile eruptions, above old and thick cratons, shields and mobile belts (Jelsma et al., 2009). As diamonds form at large depths within the mantle (>150 km) they are transported by the magma in xenocrysts and xenoliths

(Ridley, 2013). Recently, Gernon et al. (2023) correlated kimberlites with periods of dispersal of the continental plates, with a lag of about 30 My between continental break-up and kimberlite volcanism, and migration of kimberlites from the rifted margin towards the cratonic interior over time. This migration can be explained by progressive removal of the cratonic keel by convective mantle instabilities. For example, the Argyle diamond deposit in the intracontinental rift in the Halls Creek Orogen (Australia) was driven by the break-up of the Nuna supercontinent (Olierook et al., 2023).



### 4.1.2 Aggregate (construction) materials

Rift environments contain a variety of aggregate mineral resources that can serve as construction materials. For instance, deposits of sand and gravel in rift environments are of great importance for construction projects since they are used for roads and building foundations and the preparation of cement and concrete (Hearn, 2022). Clay deposits can be used for brick production, and sedimentary rocks are used for the construction and decoration of buildings. For instance, Triassic Buntsandstein outcropping on the rift shoulders of the Upper Rhine Graben has always been a popular building material in the region (Heap et al., 2017). Another application is the use of Kieselkalk, which was deposited on the European rifted margin prior to the Alpine orogeny (www.strati.ch), as railway track ballast (Suhr and Six, 2020). Volcanic deposits found in rift settings are also of interest to construction. Imported basalt blocks have been applied to reinforce waterworks in the Netherlands (Wichman et al., 2009), and intermediate lavas were used to construct the old town and iconic black cathedral of Clermont-Ferrand in the Massif Central (Dompnier et al., 2014). Eruptive materials in the East African Rift System are used for various local construction purposes, but have the drawback that their characteristics are rather unpredictable (Walle et al., 2000; Hearn, 2022). Even so, cinder material is a reasonable substitute for scarce gravel in East Africa (Hearn, 2022).

### 4.1.3 Helium gas

Another highly valuable mineral resource that can be produced in rift systems is helium gas, a crucial cooling agent (Montoya et al., 2019). A highly strategic non-renewable resource, helium is normally extracted from rocks in felsic cratons, which accumulate the gas that is generated as a by-product of uranium and thorium decay (Hand, 2016). However, unusually high helium fluxes in the Tanzanian part of the East African Rift System have recently caught the attention of exploration geologists, and large helium reservoirs have been discovered (Hand, 2016). The cause of these high fluxes is the increased heat of the rift system, which releases the helium from cratonic rocks, after which it accumulates in porous reservoir rocks from which it can readily be produced (Hand, 2016).

## 4.2 Geo-energy resources

Geo-energy resources come in various forms. Most well-known are hydrocarbons or fossil fuels (petroleum, natural gas and coal), which have fueled our economies since the industrial revolution and allow our present-day living standards. However, the use of hydrocarbons has caused massive greenhouse gas emissions, leading to increasingly severe climate change (e.g., IPCC, 2023). Consequently, one of the biggest challenges for the energy transition will be to find new energy sources to replace hydrocarbons and reduce greenhouse gas emissions. Two promising geo-energy sources that can be produced in rift environments are natural hydrogen gas ($H_2$), and geothermal energy.

### 4.2.1 Hydrocarbons and fossil fuels

Hydrocarbons presently represent the most well-known energy resources produced in rift systems, with massive oil and natural gas deposits found in (conventional) petroleum provinces such as those in the North Sea rift and UK-Norwegian margin, and



in the Gulf of Mexico (e.g., Levell and Bowman, 2018; Snedden and Galloway, 2019, Fig. 12). Rift systems provide ideal environments for the development of conventional petroleum systems. Firstly, they allow for the development of hydrocarbon source rocks, typically fine-grained marine or lacustrine sediments rich in organic material, in restricted rift basins with fast subsidence, limited water circulation and low $O_2$ content, so that the organic matter is preserved and only minor amounts of

other sediments are deposited in these sediment-starved environments (Katz, 1995; Nemčok, 2016). Rapid subsidence in rift basins also allows the source rock to be buried to a depth of 2-7 km depth by subsequent sedimentary infill to reach temperatures of ca. 100-250°C that allow for the generation of oil and natural gas (Nemčok, 2016). Furthermore, the large variety of deposits in rift systems ensures that there is a high likelihood that reservoir rocks are available, especially in earlier stages and near active normal faults (Fig. 7, White et al., 1999; Gawthorpe and Leeder, 2000). The upward migration of bouyant hydrocarbons from

the source rock into reservoir rocks can occur vertically, but pathways such as and high-permeability sedimentary rock layers and normal faults in rift systems, especially where individual rift basins merge, allow for enhanced hydrocarbon migration (Fossen et al., 2010). The presence of impermeable rock layers such as clays (or evaporites) that generally develop in the later stages of rifting act as seals or caprocks, forcing hydrocarbons to migrate around them (de Jager and Geluk, 2007). When this is however not possible, as in the case of a tectonically or stratigraphically formed trap structure, a hydrocarbon field can develop,

which can be drilled and exploited (Nemčok, 2016). It is key that these elements need to be present at the right time and the right place, so that a thorough understanding of a basin's geological history is required to successfully exploit its conventional petroleum system(s) (Magoon and Dow, 1994; Alves et al., 2020).

Further, relatively recent technological developments have allowed the exploitation of unconventional petroleum systems, which include shale oil and/or shale gas, and are much simpler than conventional petroleum systems since the source rock is

simultaneously the reservoir rock (Muther et al., 2022). Large volumes of hydrocarbons remain trapped in the impermeable fine-grained source rocks themselves, and can be released and produced by directly drilling into the source rocks followed by hydraulic stimulation, or "fracking" (Bažant et al., 2014; Li et al., 2015). Rift settings provide an ideal environment for the development of source rocks, which can be exploited even when no suitable conventional reservoirs, traps or seals are present in the system. Such unconventional production of shale oil and gas has been booming in the USA since the late 20[th] century,

and various rift basins producing unconventional oil and gas (e.g., the Southern Oklahoma Aulacogen, (Keller and Stephenson, 2007; van der Elst et al., 2013). Although there seems to be good potential in other rift basins elsewhere around the globe (e.g., the North Sea rifts, the Pannonian Basin and Donetsk Basin in Europe or the Songliao, Fuxin, Bohai Basin in China; Schulz et al., 2010; Zhang et al., 2023, respectively), these resources have not been developed as of yet. A notable exception is the Neuquen Basin in Argentina, which contains the highly prolific Vaca Muerta source rock that was deposited in a back-arc rift

setting (Howell et al., 2005).

Gas hydrates represent another type of non-conventional hydrocarbon resource that is abundantly found along rifted margins (Ruppel and Kessler, 2017), and uniquely in the continental Baikal Rift as well (Khlystov et al., 2019). At great depth, or at low temperatures in more shallow marine environments such as in the Arctic, natural gas (commonly generated by biochemical activity in the seafloor) and water can form an ice-like compound referred to as gas hydrate (Ruppel and Kessler, 2017).





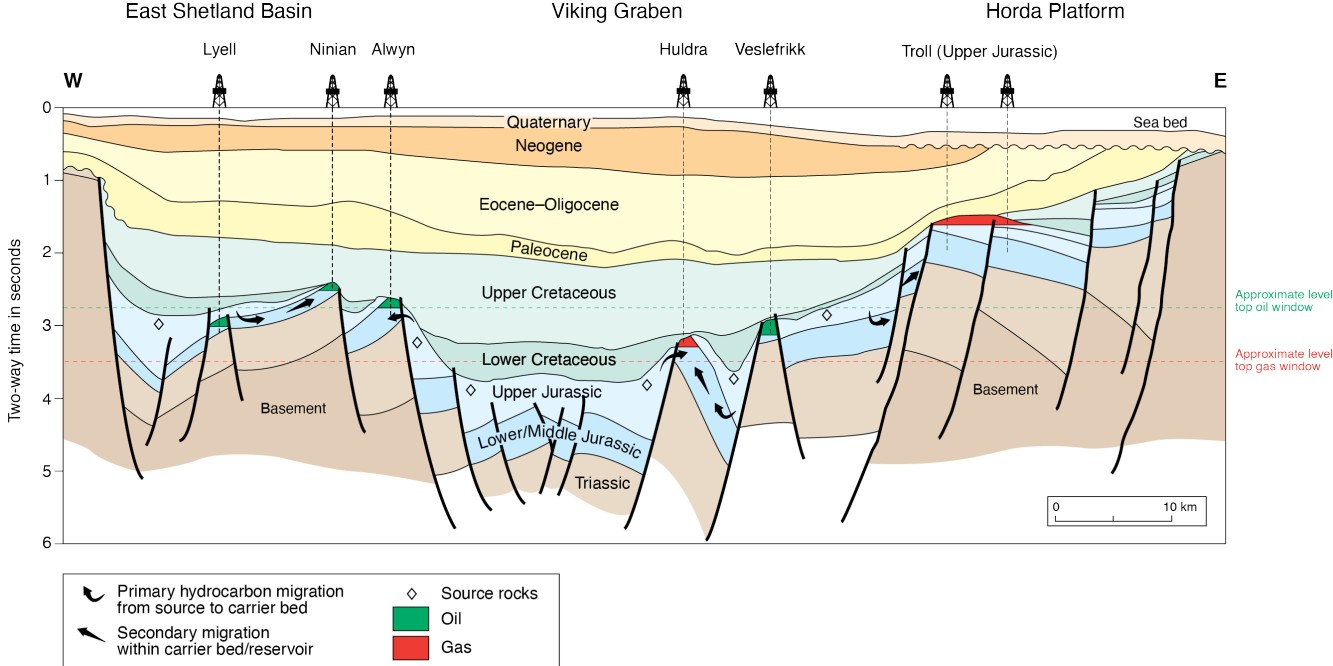

**Figure 12.** Petroleum system in the northern part of the Mesozoic North Sea Rift System. Charge is from the synrift Upper Jurassic source rocks, migration follows faults and permeable layers, the main reservoir/seal pair is the Middle Jurassic prerift Brent Formation, the traps are tilted footwall closures below the overlying Lower Cretaceous shales. After Husmo et al. (2003).

Massive deposits of gas hydrates are known to exist along rifted margins, and various pilot projects have been undertaken to explore its potential (e.g., at the margin of the Canadian Arctic and the South China Sea, Yamamoto et al., 2022).

Finally, hydrocarbons also come in solid form, i.e. as (brown) coal, or its predecessor peat. Not only does coal generate natural gas that can accumulate in hydrocarbon reservoirs, it can also be directly exploited, similar to unconventional shale gas source rocks (coal-bed methane, Moore, 2012; Muther et al., 2022). Similarly, deposits of the more advanced forms of

coal (lignite and eventually anthracite, forming after increased burial over time) have been mined for a long time, fueled the industrial revolution, and remain an indispensable energy source for many countries around the world to this day (Pudasainee et al., 2020).

### 4.2.2 Natural hydrogen (H$_2$)

Hydrogen gas (H$_2$) is a clean source of energy since H$_2$ combustion generates nothing but water as a by-product. The problem

is however that present-day H$_2$ production is costly at best (when using green energy), or highly polluting at worst (when using fossil energy) (e.g., Ajanovic et al., 2022; Osman et al., 2022). However, various sources of naturally occuring H$_2$ exist, of which the most promising is the serpentinization of (ultra)mafic rocks (e.g., mantle rocks): by reacting with water, these mantle rocks release natural H$_2$ (Fig. 13) (e.g., Smith et al., 2005; Gaucher, 2020). Large amounts of natural H$_2$ are released during



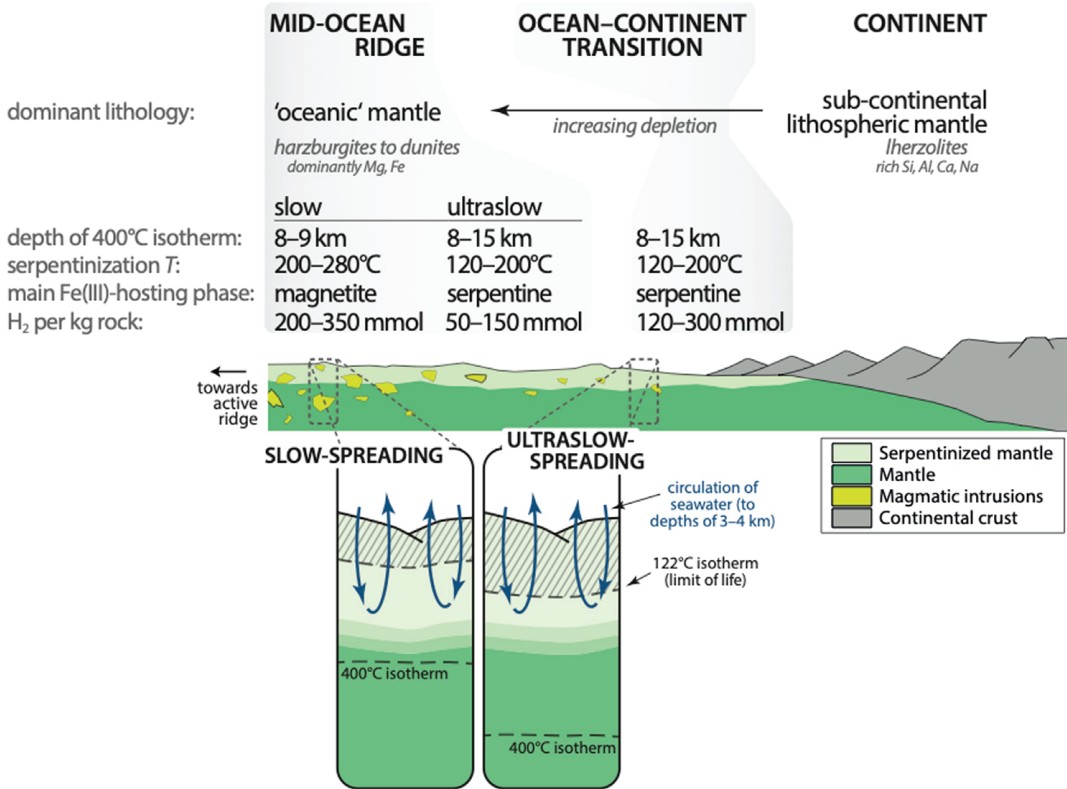

**Figure 13.** Sketch of natural $H_2$ generation related to serpentinisation at (magma-poor) rifted margins, as well as ultraslow- and slow-spreading mid-oceanic ridges. Differences in $H_2$ generation potential are due to petrological variations in the serpentinising rocks, as well as due to changes in the thermal regime, which itself strongly depends on the divergence velocity.

the more advanced stages of rifting (i.e., break-up and drifting, when mantle material is being exhumed and serpentinised)
(e.g., Albers et al., 2021; Liu et al., 2023, Fig. 13), and it is speculated that such natural $H_2$ could have played a key role in the emergence of life on Earth (Russell et al., 2010). Of key importance is that water can reach down to the mantle material, for instance via large normal faults, but complex and deep-rooted fault patterns in rift transfer and transform zones provide improved opportunities for water circulation, as is also the case for magma and hydrocarbon migration (see sections 3.2 and 4.2.1). Similar to conventional petroleum systems, the generated natural $H_2$ from such "hydrogen systems" will have to migrate
from the (mantle) source rock to reservoirs in order to be exploited, or it may even be possible to directly drill into the mantle and hydraulically stimulate the mantle source rock in order to serpentinise it (Lefeuvre et al., 2022; Zwaan et al., 2023). In the case of a natural $H_2$ reservoir, a constant influx of natural $H_2$ may be required, as the small size of the $H_2$ molecule means that it can readily escape the most impermeable seal rocks (Muhammed et al., 2023). Furthermore, $H_2$ is highly reactive and can be lost in various (bio)chemical processes. Therefore, reservoirs will ideally have temperatures between 100-200°C, at which $H_2$
is relatively inert (Lefeuvre et al., 2022).





### 4.2.3 Geothermal energy

Finally, geothermal energy production is a developing industry in the recent (continental) rift basins around the world (e.g., Kölbel et al., 2023). The thinning of the lithosphere and the rise of hot mantle material towards the surface creates an elevated geothermal gradient and the resulting higher heat flow can be exploited. Geothermal springs in rift basins such as the Upper Rhine Graben have been in use for bathing and healing purpose since at least Roman times (e.g., Sanner, 2000). Furthermore, warm water from shallow aquifers (low-enthalpy systems) (Lee, 2001) can be used for heating (green)houses and other buildings (Aydin and Merey, 2021), whereas water from deep aquifers is hot enough for power production (high-enthalpy systems, Stober and Bucher, 2021). The drilling targets for geothermal power production pose a challenge, as they may be situated in the deepest part of the rift basins, in contrast to drilling targets for hydrocarbon production that tend to occur higher up in the basin stratigraphy (Weert et al., 2023). However, geothermal energy production also provides additional mining opportunities, by extracting dissolved elements and minerals from the geothermal fluids (e.g., Lithium, Rare Earths, salts; Kölbel et al., 2023). Various geothermal projects are currently underway in continental rifts (e.g., in Soulz-Sous-Forêts, France, and Kenya; Ledésert et al., 2021; IRENA, 2020), with the East African Rift as an obvious target (Martin-Jones et al., 2020). However, subareal oceanic spreading ridges in Iceland and Afar provide perhaps the greatest potential. In the Icelandic case, the country has such a massive surplus of geothermal energy that energy-intensive industries such as aluminium production have set up camp (Leifsson, 1992). The potential in Afar remains untapped as of now, but its geothermal situation, which is similar to that of Iceland, could provide the surrounding regions with copious amounts of green energy in the future (Cherkose et al., 2023).

### 4.3 Fresh water and fertile soils

Next to mineral and energy resources, rifts environments also provide geo-resources that are crucial to sustain life, such as water and soils. Fresh water is vital to human survival, but is also a indispensable resource for many industries. In rift systems, meteoric water precipitates on rift shoulders or uplifted rifted margins such as the Ethiopian plateau, where two-thirds of the Nile water originates (Pacini and Harper, 2016), and for instance the coastal ranges of western India (Sharma et al., 2022) or Norway (Maystrenko et al., 2020). Infiltrating meteoric water can accumulate in porous rock layers (aquifers), from which it can be produced by drilling wells (e.g., the Kobo Girana Valley Development Program in Ethiopia; Zwaan et al., 2020). Fresh water also accumulates in continental rift basins, where lakes such such as Lake Baikal in Siberia and the Great Lakes of the East African Rift System represent important reservoirs. Furthermore, offshore fresh, or relatively freshened, groundwater has recently been identified as a potentially vast resource with a global volume of 1 million cubic kilometers. Such offshore freshened groundwater predominantly occurs within 55 km of the coastline and is thought to have been primarily emplaced during Pleistocene sea level low-stands (Micallef et al., 2021). By contrast, some lakes found in rift settings are highly saline, due to excessive evaporation and/or the presence of large salt deposits in their subsurface (e.g., Lake Afrera in the Afar rift), which provide opportunities for wellness activities, salt production and rare element extraction (e.g., Varet, 2018, see also section 4.1.1).



Highly fertile soils are another key resource that is abundant in rift environments. Such soils may come in the shape of the large amounts of (fine) sediment deposits, especially in areas with large rivers and large delta's. For example, the yearly
flooding of the Nile transports large volumes of silt from the Ethiopian highlands downstream to Sudan and Egypt (Fielding et al., 2018). Individual (continental) rift basins can also accumulate large volumes of fertile sediments (e.g., along the East African Rift System, or the European Cenozoic Rift System). In the case of the Nile, the silt that is dominantly brought in from the Ethiopian highlands is extremely fertile due to its volcanic origin, since these highlands are covered by up to 2 km-thick basaltic layers (Fielding et al., 2018). The presence of such rift-related volcanic rocks and soils also allows for extensive
agriculture in Ethiopia, supporting 10s of millions of Ethiopians living in the highlands (Hurni et al., 2010). Further south, the Eastern Branch of the East African Rift System Is highly prone to volcanism, providing large extents of fertile soils, and similar fertile soils are also found, but to a lesser degree along the less volcanic Western Branch of the same rift system (Ebinger, 1989).

## 4.4 Geological storage

In the effort to achieve energy secure net-zero societies, geological storage is important for a number of reasons. First, geo-
logical storage of fuels (e.g., hydrocarbons, hydrogen gas) allows the medium-to-long-term accumulation of large volumes of energy-dense fluids far from their location of production and near to where they will be required, improving energy security amongst seasonal changes in energy demand and uncertainties within the international energy supply chain. Moreover, the geological storage of energy (fuels, compressed air, heat) over short-to-medium terms aids in balancing the significant unpredictability and seasonal fluctuations inherent to renewable energy sources (e.g., wind, solar), allowing these energy sources
to meet the also fluctuating demands of society (Mitali et al., 2022). Second, geological storage provides a means to safely dispose of waste products, e.g., carbon dioxide, which require long-term isolation from the atmosphere and/or biosphere.

### 4.4.1 Temporary storage

The best known temporary geological storage targets are old hydrocarbon fields and porous rock layers, which are plentiful in rift systems. For instance in Western Europe, excellent reservoirs provide the temporary storage of natural gas that is imported
from other regions of the world (Tarkowski et al., 2021), smoothing supply to meet seasonal changes in demand and improving energy security. Porous rock layers also provide opportunities for Aquifer Thermal Energy Storage (ATES). Through ATES, heat can be transfered to water and injected into relatively shallow reservoir layers, where the heat can be effectively stored due to decreased heat loss due to advection (water flow) and diffusion (Fleuchaus et al., 2018). Of key importance for storage of natural gas and heat in subsurface reservoirs is their widespread availability and excellent permeability, which allows the
gas or hot water to be easily injected and extracted (Tarkowski et al., 2021).

Another storage target in rift systems involves salt caverns, created by dissolution mining of salt in salt diapirs, or by physical mining of evaporite layers (Duffy et al., 2023; Williams et al., 2022, Fig. 14). In contrast to porous rocks, which provide storage space between (sedimentary) grains, caverns and mine shafts are wide open spaces in the subsurface, which allows rapid injection and extraction of liquid or gaseous resources. Furthermore, salt is highly impermeable and self-sealing,
and such caverns provide ideal storing opportunities for strategic petroleum reserves, $H_2$ gas, Helium gas and compressed air





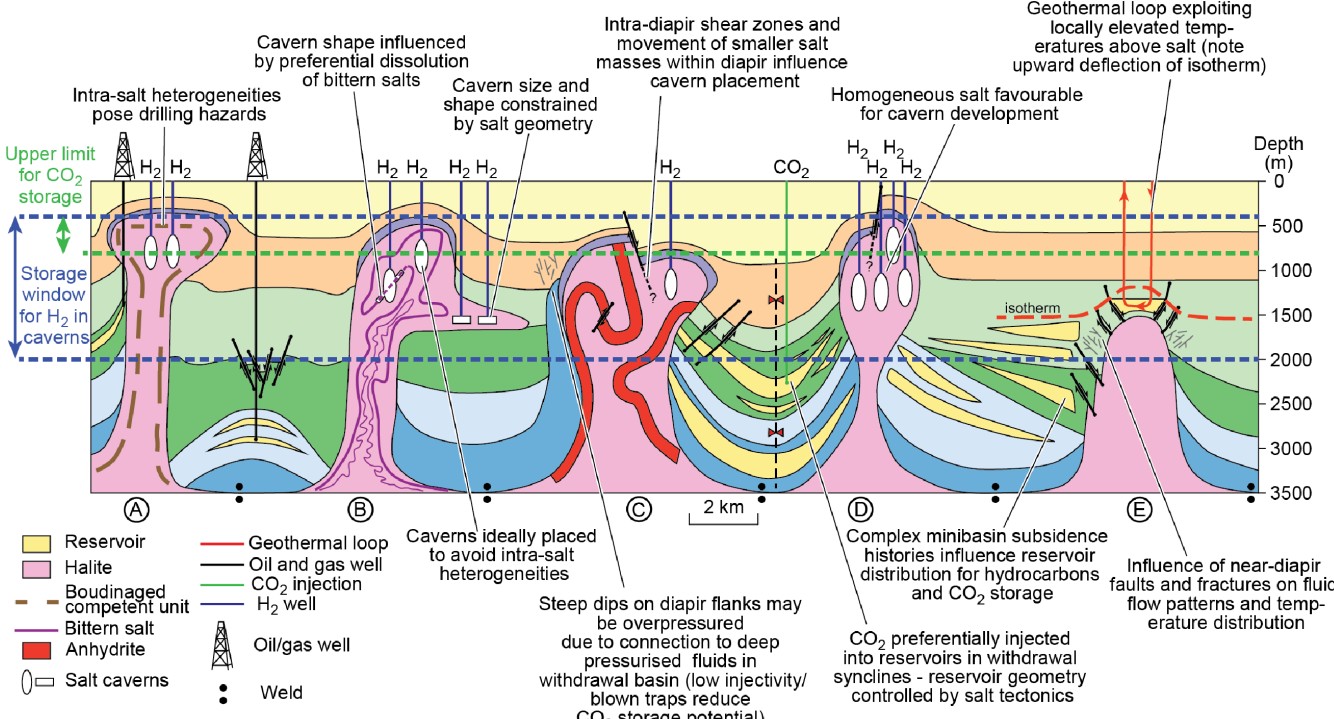

**Figure 14.** Use of subsurface evaporite (salt) structures for various geological purposes (energy storage in salt caverns, $CO_2$ storage, geothermal energy, oil and gas exploration). The great variety of salt diapir geometries and inter-diapir architecture of sedimentary rocks highlight that each setting is unique and requires a thorough understanding of its genetic processes in order to be successfully used. Adopted from Duffy et al. (2023)

(Duffy et al., 2023). Various such storage sites have been developed in the rifted margin of the USA Gulf of Mexico coast, which contains a large salt-tectonic system (e.g., Sobolik et al., 2019; Tarkowski, 2019). In particular, three commercial Compressed Air Energy Storage (CAES) facilities currently exist in Germany, the USA, and Canada, each exploiting salt caverns (Kim et al., 2023). A limiting factor for the use of salt caverns for CAES is the distribution and accessibility of suitable salt deposits, although recent work has shown that compressed air may also be stored in porous media with capacities suitable for national grid scale energy storage and efficiencies of up to 0.67 (Gasanzade et al., 2023).

### 4.4.2 Permanent storage

In our quest to establish a sustainable economy, we must not only apply the resources available to us, but also responsibly process waste products. Where some of these waste products can be recycled, some cannot and need to be permanently and safely stored, which can be done by making use of geological formations found in rift settings.

The most well-known waste product generated by industrial activity is $CO_2$, which is generally released into the atmosphere, where it subsequently is one of the principal causes of global warming. As such, it is of essential to reduce the atmospheric $CO_2$





content by capturing and permanently storing it, a process commonly referred to as carbon capture and sequestration CCS)
(Tucker, 2018; Bajpai et al., 2022). For instance, $CO_2$ can be stored in saline aquifers, or old hydrocarbon reservoirs in rift
environments (Ajayi et al., 2019), where it is simply pumped into the reservoir to take the place of the previous hydrocarbons
(Tucker, 2018). The possibilities of applying such $CO_2$ storage in depleted hydrocarbon fields in for instance the North Sea
rift and along the rifted margins of the Gulf of Mexico, China and Australia are being studied (Agartan et al., 2018; Bajpai
et al., 2022), with a successful CCS project being under way in the Danish North Sea (Skopljak, 2023). However, the storage of
$CO_2$ in depleted hydrocarbon fields comes with a number of potential challenges, including leakage of the buoyant $CO_2$ due to
abandoned infrastructure (e.g., boreholes), and human induced seismicity during injection/extraction (e.g., Ajayi et al., 2019).

In some active hydrocarbon fields, $CO_2$ is also being pumped into the reservoir to increase its internal pressure, thus enhanc-
ing hydrocarbon production while storing $CO_2$ (Whittaker et al., 2011; Bajpai et al., 2022), a process referred to as enhanced
oil recovery (EOR) or carbon capture utilisation and sequestration (CCUS). Furthermore, $CO_2$ can also be permanently stored
in sedimentary formations with poor permeability such as clay- or evaporite-rich deposits, i.e. the opposite of depleted hydro-
carbon reservoirs (Duffy et al., 2023, Fig. 14). The advantage of injecting $CO_2$ in such formations is that no reliable cap rock
is needed as in a conventional reservoir, but, akin to the source rock in unconventional hydrocarbon systems (see section 5.2),
the impermeable formation will act as both the reservoir and seal for the injected $CO_2$.

Next to permanent storage in depleted porous media reservoirs and impermeable sedimentary strata, $CO_2$ can also be chem-
ically stored when it reacts with freshly exhumed rocks, a process known as carbonation (e.g., Kellogg et al., 2019). As such,
researchers have proposed a link between global phases of plate convergence-driven mountain building and cooling climates
(e.g., Kellogg et al., 2019, Fig. 1). Fresh crustal rocks can also be exhumed during plate divergence, e.g., on the rift shoulders
of continental rift systems, leading to natural carbonation and storage of atmospheric $CO_2$. Even so, carbonation is much more
efficient when mafic rocks are available, such as mantle rocks or basalts (Matter and Kelemen, 2009). Fresh mantle rocks may
be exhumed along the rift axis or spreading ridge During the break-up stage in magma-poor systems, as well as during the
drifting stage in slow spreading systems (e.g., Albers et al., 2021; Liu et al., 2023). $CO_2$ could be artificially injected into these
exhumed mantle rocks for storage via enhanced carbonation (e.g., Olajire, 2013; Snæbjörnsdóttir et al., 2020). In magma-
rich systems, the flood basalts that erupted in continental settings, or the extensive basalt flows forming the seaward-dipping
reflectors during the break-up stage, could similarly be injected with $CO_2$ for permanent storage (Fedorik et al., 2023). The
advantage of injection in mafic rocks over porous media reservoirs is the enhanced potential for mineral trapping through so
that the $CO_2$ will be permanently fixed to the host rock (Matter and Kelemen, 2009). Projects to explore the possibilities of
injecting $CO_2$ into basaltic rocks are under way for instance in Iceland (CarbFix project; Snæbjörnsdóttir et al., 2020) and the
Red Sea, offshore Saudi Arabia (Fedorik et al., 2023).





## 5  Future challenges and opportunities

### 5.1  Fundamental research

Rift research has come a long way since the days cartographers first remarked the coastlines of South America and Africa would fit together remarkably well Romm (1994) Our understanding has evolved from the concept of continental drift, via the recognition of oceanic spreading and the occurrence of plate subduction to the present-day concept of plate tectonics. However, there are various important scientific questions associated with rifting that remain to be answered. A recent white paper by Peron-Pinvidic et al. (2019) sums up a number of key topics to which these questions are linked, and which should be

considered in a 3D framework to account for the numerous lateral variations in rift systems (section 2.3), rather than the more traditional but limited 2D view:

**1. Rheology:** various factors impact the rheology of the lithosphere during rifting, leading to a wide range of rift architectures (e.g., Sapin et al., 2021), but we need to better constrain the timing and interaction between these factors.

**2. Inheritance:** inheritance is known to influence the localization of deformation during rifting, but the exact influence of the

different types of inheritance is challenging to entangle. Furthermore, rift inheritance can also affect subsequent contractional deformation (i.e. the "other half" of the Wilson cycle, Fig. 1), which remains poorly understood.

**3. Faults and deformation:** Rifting is generally accommodated by the development of faults and shear zones, but a detailed understanding of early fault evolution remains elusive (e.g., Rotevatn et al., 2019).

**4. Stratigraphy:** how do sedimentary processes react to tectonic deformation, and vice versa, and how well do classical

models fit with new insights (e.g., Masini et al., 2013).

**5. Kinematics:** rift systems can form under a wide range of plate kinematic conditions, we need to better understand what the drivers behind rift kinematics are, and how these kinematics affect the evolution of rift systems (e.g., Brune et al., 2016).

**6. Mantle:** Much research has focussed on the crust and lithosphere, which are easier to access and study, but the influence of the mantle on rifting remains elusive although tectonic modelling suggests it can have a dominant impact (e.g., Chenin et al.,

2015; Zwaan et al., 2022), and more research is dearly needed.

To approach these research topics, existing and novel methods need to be applied. Fieldwork in rift systems remains a key means of acquiring data, but its effectiveness can be greatly enhanced by the use of drones and detailed satellite imagery. Such satellite imagery (e.g., topography data) for instance allows for automatic interpretation using state-of-the-art algorithms and artificial intelligence (AI) to generate fault maps (Gayrin et al., 2023). Machine learning and AI in general will expand our

research options in many different ways (Chen et al., 2023). Even so, such advanced methods will rely on detailed and correct data from the real world, which remains a challenge. For instance, satellite imagery allows for bathymetry mapping of lakes and seas, which cover large parts of the world's rift systems, yet high-resolution bathymetric mapping of lake- or seafloors can only be obtained via sonar surveys during scientific cruises (Wölfl et al., 2019). Such cruises will also remain crucial for sampling the deep through seafloor dragging or the acquisition of seismic datasets.

Offshore and Onshore drilling of the lithosphere in rift settings, as done by the IODP (International Oceanic Drilling Program) and ICDP (International Continental Drilling Program), respectively, continue to draw new challenges and targets (Kop-





pers and Coggon, 2020), providing new insights and research opportunities (e.g., the planned ADD-ON project to drill into the Afar rift: ADD-ON, 2023). Similarly, large-scale geophysical efforts can provide the detailed data that allow us to better understand the subsurface geology in rift systems. Furthermore, a wealth of otherwise undisclosed geological information such

as borehole logs and seismic surveys can also be obtained from industry partners such as energy companies, mining operators and water production firms (Peron-Pinvidic et al., 2019). Indeed, increasing collaboration and exchange of information and ideas between industry and academia, as well as interaction with policy makers, may be the way to move science forward (Ankrah and AL-Tabbaa, 2015; Ludden, 2020) (see also sections 5.2 and 5.3).

Next to advancing data acquisition methods and increased collaboration, geological modelling approaches, either in the

laboratory (analogue) or using numerical codes, provide a unique means to better appreciate the long-term evolution of rift systems and the associated geological processes. Both approaches are rapidly developing and in the ideal case, researchers can combine them to get "the best of both worlds" (e.g., Brune et al., 2017; Maestrelli et al., 2022). Such interdisciplinary research, also beyond the modelling domain, poses great opportunities for advancing our knowledge of rift systems.

A fainl exciting research direction we would like to mention is the developing field that aims at deformation on planetary

bodies. Satellite imagery and geophysical analyses done by orbital and landing probes allow us to study the planets and moons in our solar system, which in various cases contain rift-like structures. Researchers have for instance identified rifts on Mars (Hauber et al., 2010), Venus (Regorda et al., 2023) and Mercury (Watters et al., 2016). Studying such planetary rift tectonics may also provide insights into early terrestrial tectonics that operated under very different conditions than in the present day (Bradley, 2008; Capitanio et al., 2019).

**5.2 Natural hazards**

Dealing with the risks posed by natural hazards in rift systems (seismicity, volcanism and mass wasting, see section 3) requires a thorough understanding of the geological processes causing these hazards (see sections 2 and 5.1). However, in order to understand a specific hazard and the risk that it poses in a specific area, researchers and the government can profit from much more detailed study and especially monitoring approaches. In the case of those geo-hazards related to earthquakes, volcanism

and mass wasting, detailed monitoring through a combination of field observations, geophysical methods (earthquake monitoring and analysis through permanent seismic networks, as well as seismic data interpretation), satellite imagery analysis (incl. INSAR), and stress measurements in boreholes (see sections 3.1.2 and 3.2.2). For estimating landslide risks, also the impact of human activity (agriculture and deforestation) needs to be assessed (e.g., Depicker et al., 2021).

Installing and expanding monitoring networks in known risk areas around the globe is clearly of great societal interest, but

another important challenge for the future may be the assessment of hazards in less obvious locations. This is especially relevant when it comes to intra-plate earthquakes along "passive" rifted margins or in old and (supposedly) tectonically inactive rift basins, or when earthquake cycles or volcanism with very long recurrence times are involved (see section 3.1). Here, multidisciplinary approaches can help to provide the best possible risk assessment. An innovative way to expand our risk assessment capacities, in particular in poorer regions in the world, is by setting up innovative monitoring networks that involve active

participation of the local community (Citizen Science, e.g., Boudoire et al., 2022; Sekajugo et al., 2022). This is especially rel-





evant to populations living in the East African Rift System, which are projected to rise considerably for the foreseeable future (Worldometer, 2023). Furthermore, although there are distinct limitations (Mancini et al., 2022), there are great opportunities to streamline analysis and improve risk pattern recognition through machine learning algorithms and AI that can recognise patterns in earthquake catalogues (e.g., Stockman et al., 2023; Zlydenko et al., 2023).

Another key means to test risks posed by geological processes in rift systems is the detailed modelling of these processes and link them to observations from nature. For example, Corbi et al. (2019), show how machine learning can predict earthquakes in analogue models, which could potentially be used for real earthquake forecasting. Likewise, modellers can use a variety of methods to simulate volcanic processes (Poppe et al., 2022), (submarine) landslides, and associated tsunamis (e.g., Berndt et al., 2009; McFall and Fritz, 2016). The output of such interdisciplinary analyses can serve to improve the existing hazard

and risk assessments. This is especially relevant for human activities that involve subsurface fluid injection, such as geothermal energy projects and unconventional hydrocarbon production, which regularly generate seismicity (e.g., Andrés et al., 2019).

## 5.3    Geo-resources

The energy transition will require huge amounts of geo-resources, which poses significant challenges and opportunities for research and development. Exploration and production of such resources will have to be stepped up by an order of magnitude,

and new areas need to be explored to satisfy the demand posed by a growing global population that is increasing its overall level of development (e.g., IEA, 2021; UNEP, 2023). Furthermore, the European Union has expressed its intention to increase the extraction of mineral resources from within the European continent (EU, 2023), and the USA have expressed similar intentions (USDC, 2019).

     A crucial challenge to geoscientists within this context will be to improve our understanding of the processes leading to the

generation of mineral deposits in rift systems, and the development of new methods to trace down and exploit these mineral resources as efficiently and minimally invasive as possible. Many of the best accessible deposits found near the Earth's surface have long since been discovered, but a wealth of mineral deposits is expected to be found deeper in the subsurface (Arndt et al., 2017). Though technically challenging, deeper mining activities have the advantage of diminishing the impact on the landscape and environment, making such activities more sustainable (Arndt et al., 2017). A promising avenue for mineral

exploration is the "mineral system analysis" approach, somewhat similar to the analysis of petroleum systems (section 4.2.1), where the full context of mineral deposit development is being assessed, thus allowing for a better understanding of where mineral exploration should focus (e.g., Hagemann et al., 2016; Lawley et al., 2022). Furthermore, ongoing exploration of the offshore parts of rift systems will provide a whole new mining environment, although such mining is still highly challenging and may have major consequences for crucial deep sea ecosystems (Washburn et al., 2019; Kung et al., 2021). As such, here

too there are great opportunities for the development of more sustainable mining or extraction methods.

     The energy transition policies also imply a definitive move away from fossil fuels, yet hydrocarbons will remain a key part of the global energy mix for the foreseeable future (Kober et al., 2020; IEA, 2023b). Hydrocarbon exploration is likely to continue in many places (e.g., in the failed rifts crossing the African continent), but in the future, energy systems that combine the use of renewable energy and CCS techniques can render the production of hydrocarbons more efficient, sustainable, and



potentially even carbon neutral (IEA, 2023a). Offshore production of hydrocarbons in rift and rifted margin environments could also become sustainable, when combined with CCS techniques and targeting the production of natural gas (in hydrate form, see section 4.2.1) instead of the more polluting petroleum and coal.

Natural $H_2$ is a true wildcard in the energy transition. Still very much underexplored, the vast expanses of exhumed mantle rocks in advanced rift systems and passive margins imply that huge volumes of natural $H_2$ are formed during rifting (Liu et al.,
2023). Perhaps, as Gaucher (2020) argues, the coming years will see the start of a flourishing natural $H_2$ industry. However, in order to successfully develop such an industry, upcoming natural $H_2$ exploration efforts should aim at identifying the crucial aspects of potential $H_2$ systems, which are rather similar in nature to those in petroleum systems (section 4.2.2, Gaucher et al., 2023; Zwaan et al., 2023). This fact represents great opportunities since very similar exploration methods as the ones used in the petroleum industry can be applied for the development of this sustainable energy source.

Geothermal energy is an ever developing and sustainable geo-energy source with great potential as already demonstrated in for instance Iceland (section 4.2.3). However, the huge geothermal potential in the largest magma-rich continental rift system in the world, the East African Rift System (e.g., Elbarbary et al., 2022), is gathering interest, but remains mostly untapped (e.g., IRENA, 2020). Future efforts to unlock these energy resources will involve the detailed assessment of the geothermal regime and subsurface geology in the various rift basins in East Africa, with particular attention to the highly magmatic Afar
rift where very little data is available, but high heat flows are recorded (Limberger et al., 2018; IHFC, 2023). There may even be possibilities of setting up local geothermal projects that can support the development of local communities (Varet, 2018). Moreover, systematical extraction of dissolved minerals and elements from geothermal fluids provides a means to add value to geothermal operations (Kölbel et al., 2023). Geothermal plants in Iceland have also been shown to emit large volumes of natural $H_2$ during their operations, which could be captured and used as an additional green energy source (Gaucher et al.,
2023). Similar natural $H_2$ emissions have been recorded in Afar as well, providing an additional motivation to explore for resources in that specific region (Pasquet et al., 2021).

Future energy requirements will demand a large increase in temporary storage capacity (Duffy et al., 2023), and attempts to reduce the concentration of atmospheric $CO_2$), which are currently still steeply rising, will require vast expansion of permanent sequestration capacity (Tucker, 2018). Our best hope of realising these requirements is to identify the most promising sites for
either type of storage, and to expand the storage capacity of these sites as much as possible. With the caveat that both types of storage will likely have to be close to industrial centres, where resources are consumed and $CO_2$ is produced. A solution may be to focus temporary storage efforts on evaporite deposits and depleted hydrocarbon reservoirs, many of which are found in rift settings, whereas CCS could focus on the widely available exhumed mantle bodies and basaltic flows around the globe (Matter and Kelemen, 2009).

Upcoming research into water resources may target subsurface water flows from rift shoulders into rift basins, as well as offshore aquifers alonf rifted margins (Micallef et al., 2021). Knowledge of these water systems will be critical to fulfil the water needs of the ever growing populations living in rift environments, and may also be important for developing local geothermal projects (Varet, 2018). Another important research topic will be the impact of these growing populations, and the impact of the associated intensified land use, on soils in terms of their capacity to yield sufficiently large harvests to sustain

these populations. Furthermore, more intense land use, deforestation and soil degradation may also increase the risks posed by natural hazards such as landslides (Depicker et al., 2021, e.g.,).

Finally, we believe that a crucial aspect of successful future geo-resource endeavours is the efficient exchange of knowledge and expertise between researchers and industry players, as well as government agencies. Such knowledge transfer benefits all involved, as it allows researchers to better understand the geo-resource at hand, be it minerals, hydrocarbons, or natural

hydrogen, which companies can use to improve their exploration strategies, and governments can profit from the resulting economic activity. Excellent examples are the DINOloket and NLOG platforms that collect and make available data regarding the subsurface of the Netherlands (TNO, 2023a, b). The DISKOS and NPD Factpages platforms are similar efforts to make available data regarding the Norwegian subsurface (NPD, 2023a, b).

## 6 Concluding remarks

Rifting and continental break-up form a key research topic within geosciences. A thorough understanding of the processes involved, as well as of the associated natural hazards and natural resources, is of great importance to both science and society. In this review we provided an up-to-date summary of these processes, hazards, and resources. In addition to reviewing the state-of-the-art in rift research, we also discussed the key challenges for the future, and identified opportunities for research and knowledge application, where especially knowledge transfer between science, industry and government can help realise

breakthroughs. We therefore hope that this review paper will inspire future research in the field of rifting.

*Author contributions.* Conceptualization: all authors; Project administration: FZ; Visualization: PC, JP, SB; Writing – original draft preparation: all authors

*Competing interests.* No competing interests to declare

*Acknowledgements.* FZ is funded by a GFZ Discovery Fund fellowship. AG is funded by a Helmholtz Recruitment Initiative. PC is supported
by the Marie Skłodowska-Curie grant agreement No 895895, project SUBIMAP, funded by the European Union's Horizon 2020 research and innovation programme. The Open Access Publication costs of this paper were covered by the "Open Access Publikationskosten" funding programme of the Deutsche Forschungsgemeinschaft (DFG, German Research Foundation), Project Number 491075472, and the research project ASTRACAN, Ref. PID2021-123116NB-100, funded by the Ministry of Science and Innovation of Spain.



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
