# Peer review of "(D)rifting in the $21^{st}$ century: Key processes, natural hazards and geo-resources"

_EGUsphere, 2023_

## Author Response (AR1)

**Author's response EGUSPHERE-2023-2548**

**Title:** (D)rifting in the 21st century: Key processes, natural hazards and geo-resources

**Authors:** Frank Zwaan, Tiago Alves, Patricia Cadenas, Mohamed Gouiza, Jordan Phethean, Sascha Brune, and Anne Glerum

**Reviewer 1**

This is a review manuscript that comprehensively presents the geological processes that control the evolution of rift systems. Importantly, this paper also links these processes to the occurrence and evolution of natural hazards and geo-resources in rift-related systems. I therefor recommend the publication of this paper.

However, the authors state that this review paper aims to transfer knowledge between academics, industry, and government departments. Therefore, it is important to review and calibrate the statements, citations, and figures in the manuscript carefully. I suggest that the authors consider the following points to improve the manuscript.

- **Reply:** We thank the reviewer for these kind words, and for the suggestions, which we have addressed below.

**Text :**

Line 37: In fact, natural disasters in rift environments have claimed thousands of lives over the course of human history (refs).

- **Reply:** We have added some references that were already used elsewhere in the text to back up this claim, and we slightly modified the sentence to:
  - "In fact, natural disasters in rift environments have caused extensive damage and claimed many lives over the course of human history (e.g., Gouin 1979; Liu et al. 2007; Hearn 2022).

Line 61: 'Seismic datasets': Would you consider using 'geophysical datasets instead? It's important to note that, in addition to seismic data, other geophysical data such as gravity and electromagnetic data are crucial in understanding the rifting processes.

- **Reply:** The original listing of methods was not intended to be exclusive. At the same time, we believe that seismic interpretation merits a mention. To avoid any biased impression, we have modified the sentence to (bold is new text):
  - "These methods include geological mapping and sampling, borehole logging, interpretation of 2D and 3D seismic **and other regional geophysical** datasets, aerial and satellite observation, as well as analogue and numerical modelling of rifting processes."

Line 119: I think Buck 1991 should be quoted in reference to Narrow and Wide rift.

- **Reply:** We have added the reference here.

Line 133: I can understand this sentence, when combined with the previous one. But this sentence seems confusing on its own, because the deformation of the oceanic rift system does not seem to be most dominantly controlled by the rheology of the oceanic lithosphere, but rather magmatic processes are a very important controlling factor (See Buck et al., 2005; Behn and Ito, 2008)

- **Reply:** We have rephrased this sentence to the following (bold is new text):
  - "Deformation in an oceanic rift system (Drifting stage, Fig. 3) is then mostly controlled **by the degree to which extension is accommodated by magmatic intrusions (Buck et al., 2005).**
    - New citation: Buck, W. R., Lavier, L. L., & Poliakov, A. N. B. (2005). Modes of faulting at mid-ocean ridges. Nature, 434(7034), 719–723. https://doi.org/10.1038/nature03358

Line 136-139: I think this needs to be more accurate. When the rate of extension is slower, the cooling effect of the seawater circulation is able to make the material in the center of extension (deformation) that would otherwise be hotter due to mantle upwelling colder and stronger, thus preventing further concentration of deformation and the formation of wide rifts (See Brune et al., 2023).

- **Reply:** Many thanks for pointing this out, we have added some nuance to the text, it now reads as follows (bold is new text):
  - "[...] whereas faster plate motion increases coupling, promoting narrow rifting (Fig. 4). **However, feed-backs between thermal and mechanical processes render the process more complex: at low extension rate, when the rift center has sufficient time to cool either through conduction or hydrothermal circulation, rift strength may increase and eventually lead to relocalisation of the rift center or to the abandonment of the rift branch (van Wijk and Cloetingh, 2002)."**
    - New citation: van Wijk, J. W., & Cloetingh, S. A. P. L. (2002). Basin migration caused by slow lithospheric extension. Earth and Planetary Science Letters, 198(3–4), 275–288. https://doi.org/10.1016/S0012-821X(02)00560-5

Line 139-144: I think this needs to be revised. There is no evidence that the slower the rate of spreading, the weaker the lower crust is at the centre of oceanic spreading, while on the contrary, there is a lot of evidence that the slower the rate of spreading, the greater the thickness of the brittle lithosphere (e.g., Grevemeyer et al., 2019) i.e., that the lower crust is stronger (because it is colder). Also, it is not true, as the authors suggest, that the slower the rate of spreading, the easier it is to develop OCCs, and very typical OCCs are also developed at some slow ocean mid-ridges, whereas at many ultra-slow mid-ridge segments, OCCs are not developed (Cannat et al. 2006; Thoukle et al. 2008).

- **Reply:** The reviewer is right to point out that there are some errors in our text. Indeed, slower spreading rates tend to favor rift valleys, whereas faster rates lead to broad axial highs. Moreover, ultra-slow ridges tend to allow core complex formation. But these general trends are not always applicable due to the impact of magma supply, which can strongly affect the morphology of a mid-oceanic ridge as well. We have now significantly reworked the text as follows:

- During the Drifting stage, ultra-slow plate divergence (v < 10 mm/yr) can allow for oceanic core complex formation (Brun et al., 2018), for example along the SW Indian Ridge and the Gakkel Ridge in the Indian and Arctic Ocean, respectively (Dick et al., 2003; Zhou et al., 2022). Slow (10-40 mm/yr) spreading ridges can also host core complexes, but tend to develop more typical mid-oceanic ridges with well-defined axial valleys, as observed along the Mid-Atlantic Ridge (Macdonald, 2019). By contrast, faster plate motion (> 90 mm/yr) during the Drifting stage generally leads to axial highs that are elevated several 100s of meters, for instance along the mid-oceanic ridges of the Pacific (Macdonald, 2019). However, this general relation between plate divergence rate and mid-oceanic ridge morphology type does not always hold since the morphology of mid-oceanic ridges can be significantly altered depending on rates of magma supply as well (Cannat et al. 2006; Macdonald, 2019)."
  - New citation: Cannat, M., Sauter, D., Mendel, V., Ruellan, E., Okino, K., Escartin, J., Combier, V., Baala, M., (2006) Modes of seafloor generation at a melt-poor ultraslow-spreading ridge. *Geology* 34 (7): 605–608. doi: https://doi.org/10.1130/G22486.1

Line 150: should be Brune et al., 2016 ?

- **Reply:** Indeed, thanks for noticing, it is corrected.

Line 156-158: please list references.

- **Reply:** We have added three relevant references here (which were already cited elsewhere in the manuscript): Dick et al. (2003), and MacDonald et al. (2019), and a new citation: Demets et al. (2010). See also the modifications to the description of the links between plate motion and mid-oceanic ridge type that is linked to the information presented in these sentences.
- New citation: DeMets, C., Gordon, R.G., Argus, D.F. (2010) Geologically current plate motions, Geophysical Journal International, Volume 181, Issue 1, Pages 1–80, https://doi.org/10.1111/j.1365-246X.2009.04491.x

Line 267: I think it's too arbitrary to say. In addition to BDT depth, other factors such as the difference between pore pressure in the extension and compression environments are also important.

- **Reply:** Many thanks for the comment, we have rephrased the sentence as follows (bold is new text):
  - "However, in rift systems, fault planes do not reach as deep as in subduction zones because of a shallower brittle- ductile transition caused by **relatively higher temperatures, and due to other factors such as the lower pore fluid pressures in rifts**."

**Figures :**

Figure 1: I don't understand how the East Indian Ocean can belong to both the oceanic subduction phase (with oceanic spreading) and the Mature subduction phase (without oceanic spreading).

- **Reply:** Thanks for pointing this out, this seems to be a copy-paste error. We have removed the Eastern Indian Ocean from the Mature Subduction panel.

Figure 3-S5: I would suggest adding the oceanic crust to the figure.

- **Reply:** We agree, it has been added in Stage 5 of the figure.

Figure 5: I would suggest not juxtaposing tectonic extension along with magma-poor rifted margins and magma-assisted extension along with magma-rich rifted margins. It would give the reader the false impression that there is a correspondence between them. But in fact, the cartoon on the left is just a good representation of the amount of yield force required on the two different types of extension, and even with magma-rich rifted margins, most of the extension is done by tectonic extension i.e. faulting.

- **Reply:** We agree that Fig. 5 may give the wrong impression. In order to avoid any confusion, we have now moved panels (a) and (b), which show the general impact of magma-poor and magma-rich rifting during (early) rifting above panels (c) and (d), which show the characteristics of magma-poor and magma-rich margins. We also modified the figure caption that now reads as follows (bold text is new):
  - "Figure 5. **Differences between magma-rich and magma-poor systems. (a-b) Continental rifting with and without contemporaneous magmatic intrusions,** redrawn from (Buck2006). The yield stress required for extension to progress is significantly lower where magmatic injections weaken the lithosphere. (c-d) Characteristics of magma-poor and magma-rich rifted margins, which form during **advanced rifting (break-up) with and without** contemporaneous magmatism. **Modified after Franke (2013).**"

Figure 9: It seems that the legends for 'Volcano' and 'Large Igneous Province' are incorrect.

**Reply:** Thanks for noticing this error, it has been corrected.

**Reviewer 2**

It was a real pleasure to read this review manuscript dealing with geological processes acting within the rifts systems and their implication regarding natural hazards and geo-resources.

I found the manuscript very interesting, synthesizing the overall knowledge the scientific community brought in the last century or so.

I would recommend this manuscript to be accepted after minor revisions, I specify my comments below.

- **Reply:** We thank the reviewer for these kind words, and for the suggestions, which we have addressed below.

**Comments:**

- there is almost no discussion about the climate (and its variations) impact on rift systems, specifically on its role on denudation rate of rift shoulders for instance, or sea-level fluctuations within rift basin and their sequences (subsidence and compaction effect)

  o **Reply:** Many thanks for raising this point. We have added several sentences to address it, but we could not expand too much on the climate impact (similar to many other interesting topics), to avoid making the paper too long (new [draft] text in bold):

- **"Finally, climate is known to influence rift systems by controlling the rates of denudation and erosion of evolving basins in time and space (Friedmann and Burbank, 1995; Leeder et al., 2008; Salgado et al., 2016; McNeill et al., 2019). The most pronounced effect of climate on rift basins is often recorded in the type and volume of sediment deposited in the basin per se, as depositional facies and sediment influx rates vary dramatically with changing environmental conditions. The best example of this is, perhaps, the onset of microbial carbonate deposition in hypersaline basins, subject to extreme evaporation conditions, in the South Atlantic Ocean during Break-up (Alves et al., 2020). However, to discern tectonic 'pulses' from climatic ones can be challenging as the rift basin, sea/base level and the surrounding topography, are naturally active and variable (Gallagher et al., 1999). Hence, climate change can usually be identified when analysing 1st and 2nd order depositional sequences from different rift basins across the world, but is not resolved easily at the 3rd order due to the potential impact of global events suppressing local climatic signals (e.g., Mackay, 2007; Mazzini et al., 2023)."**

    ▪ New references:
      ▪ Alves, T.M., Fetter, M., Busby, C., Gontijo, R., Cunha, T.A., Mattos, N.H. (2020). A tectono-stratigraphic review of continental breakup

    on intraplate continental margins and its impact on resultant hydrocarbon systems. Marine and Petroleum Geology 117, 104341. https://doi.org/10.1016/j.marpetgeo.2020.104341

- Friedmann, S.J. and Burbank, D.W. (1995). Rift basins and supradetachment basins: intracontinental extensional end-members. Basin Research, 7, 109-127.
- Gallagher, K. Brown, R., Osmaston, M., Ebinger, C., Bishop, P. (1999). Denudation and Uplift at Passive Margins: The Record on the Atlantic Margin of Southern Africa. Philosophical Transactions: Mathematical, Physical and Engineering Sciences, Vol. 357, No. 1753, Response of the Earth's Lithosphere to Extension, pp. 835-859.
- Leeder, M.R., Harris, T., Kirby, M.J. (2008). Sediment supply and climate change: implications for basin stratigraphy. Basin Research, 10, 7-18. https://doi.org/10.1046/j.1365-2117.1998.00054.x
- Mackay, A. W. (2007). The paleoclimatology of Lake Baikal: A diatom synthesis and prospectus. Earth-Science Reviews, 82, 181-215. https://doi.org/10.1016/j.earscirev.2007.03.002
- Mazzini, I., Cronin, T.M., Gawthrope, R.L., Collier, R.E.Ll. et al. (2023). A new deglacial climate and sea-level record from 20 to 8 ka from IODP381 site M0080, Alkyonides Gulf, eastern Mediterranean. Quaternary Science Reviews, 313, 108192. https://doi.org/10.1016/j.quascirev.2023.108192
- McNeill, L.C., Shillington, D.J., Carter, G.D.O. et al. (2019). High-resolution record reveals climate-driven environmental and sedimentary changes in an active rift. Sci Rep 9, 3116. https://doi.org/10.1038/s41598-019-40022-w
- Salgado, A.A.R., Rezende, E.A., Bourlès D. et al. (2016). Relief evolution of the Continental Rift of Southeast Brazil revealed by in situ-produced 10Be concentrations in river-borne sediments. Journal of South American Earth Sciences, 67, 89-99. https://doi.org/10.1016/j.jsames.2016.02.002

- line 95: due TO mantle processes...

- **Reply:** Thank you for noticing, we have added "to".

- lines 135-150: when discussing fast and slow plate motions, it would be great to provide the reader with estimations

- **Reply:** In the text we refer to high and low strain rates, which are related to fast and low plate velocities, but the expression of strain rate/plate velocity depends on many additional factors (temperature, lithospheric thickness/presence of ductile layers]. Moreover, there is a clear classification for oceanic rifting/spreading (anything below 40 mm/yr is slow, which is linked to specific mid-oceanic ridge morphology), whereas slow continental rifting may be < 5 mm/yr, but which does not necessarily defines the structural style of the rifts. As such, we refrain from providing exact plate motion rates for continental systems in light of the impact of strain rate on rift style, but specify the < 40 mm/yr rate as a slow rifting rate for oceanic systems.
    - Further in the text we do mention that continental rifting velocities typically fall in the range of 1-20 mm/yr.

- line 230: do all rift basins follow an overall transgression trend? maybe at the beginning but they rather all follow a transgressive-regressive trend every subsidence pulse (see. Martins-Neto and Catuneanu, 2010).

- **Reply:** It is true that there is some more nuance to this overall statement, but the general trend in developing rift systems is one of transgression. We we have now included some details, as follows (new text is bold):
  - "The stage of rifting, location in the rift basin and the provenance of the sediments largely determine the sedimentary infill of a given site, which, nevertheless, **despite minor regressive episodes during periods of falling sea level or high relative sedimentation-subsidence rates (Martins-Neto and Catuneanu, 2010),** follow a large-scale transgressional sequence as a rift system evolves."
    - New citation: Martins-Neto, M.A., Catuneanu, O., Rift sequence stratigraphy, Marine and Petroleum Geology 27, Pages 247-253. https://doi.org/10.1016/j.marpetgeo.2009.08.001

- line 301: or the instead of "orthe"

- **Reply:** Thanks for noticing, it is corrected.

- line 330: EARS is cited without the acronym definition.

- **Reply:** Thanks for noting, we have now opted to spell out "East African Rift System" throughout the text to avoid any confusion with the acronym.

- line 374: I think a verb is missing on the sentence "However, the last eruption in the Massif Central only around 8 ka"

- **Reply:** indeed, "occurred" is missing here, and is added.

- line 584: buoyant instead of bouyant

- **Reply:** it is corrected.

- line 600: there is an extra ( before Keller

- **Reply:** thanks for noting, we have corrected it.

- line 746: please add a point after Romm (1994)

- **Reply:** it is done.

- Fig. 1: there is no legend for the astenospheric mantle

- **Reply:** Similar to the original figure in Wilson et al. (2019), we have omitted adding the asthenosphere. Tests with the asthenosphere included made the figure rather unappealing. As such, we prefer to keep the asthenosphere out, but have specified this in the caption to avoid confusion, by adding the following statement:.
  - **"Note that the asthenosphere below the lithospheric mantle is not depicted"**

- Fig. 8: is there a possibility to color-code earthquakes given their hypocentre depth? I'm asking because I'm wondering if comparing divergent rift-related earthquakes within the East African plate is coherent with convergent rift-related earthquake within the Mediterranean domain

- **Reply:** Thanks for that suggestion. We have tried several options, but color-coding by depth won't solve the problem to differentiate between rift and collision related earthquakes. We have used previous compilations to indicate the location of major earthquakes in extensional settings by white and red colors (where the red ones are explicitly mentioned in the text).

- Fig. 9: some of the volcanism depicted on the picture is related to convergent settings (subduction zone), should this volcanism be compared to rift-related volcanism? any way to decipher those types on the figure?

**Reply:** We have updated the legend and added locations mentioned in the text. This should guide the eye to those places where volcanoes are related to divergent plate boundaries.

---

## Author Response (AR2)

**Reply corrections:**

NB: Line numbers are according to the previous manuscript version

*Figure 2. I cannot find "Top" figure.*

- Thanks for noticing, we have removed "Top" from the figure caption

*Figure 3. In S3, where are the normal faults generated in S2? I also find color difference of asthenosphere between S5 and the legend.*

- We have drawn in abandoned faults (as faint dotted lines), and checked the colours. There should be no difference in asthenosphere colour between figure and legend.

*Line 88-89: I can understand sorting rifting by active and passive, but what about back-arc rifting and orogenic? Are they passive or active? Or do they belong to new types?*

- We see that the text is a bit confusing as these definitions are somewhat overlapping, and have rephrased things for clarity:
  - "It must be noted that we consider these five stages to be representative of the general evolution of rift systems, independent of the tectonic environment in which they develop; rifting may be initiated by asthenospheric upwelling (active rifting) due to for example mantle plume activity or slab break-off, or by far-field stresses (passive rifting) due to for example subduction rollback or, continental collision. These different causes leading to rifting can overlap in space and time, and have inspired multiple rift classifications in the literature (e.g., Merle, 2011; Şengör, 2020; Peron-Pinvidic, 2022), yet the expression of rifting processes generally follows the same general trend as outlined above."

*Line 131: Delete "from occurring"*

- The issue seems to be the use of "avoid" in "although magmatism can avoid this shift from occurring", so we have slightly rephrased the last part of this sentence, which now reads as follows:
  - "although magmatism can **prevent** this shift from occurring."

*Line 144: Only half bracket before v.*

- Thanks for noticing, we have added the missing bracket.

*Line 173: Localisation. Be consistent with British.*

- We have corrected it here, and in a couple other instances in the text.

*Line 191-194: This sentence is too segmented. Better rephrased.*

- We have shortened the sentence for better clarity. It now reads as follows:

- o "As a consequence, the structural inheritance the individual rift segments follow can result in different structural styles along each rift segment, ranging from orthogonal rifts to oblique and even transform systems (e.g., Corti et al., 2007; Agostini et al., 2011)."

*Figure 5-legends: Underplating of what?*

- In the light of current uncertainty if this is in fact "Magmatic underplating", we have now changed things to "High-velocity materials (Vp > 7.2 km/s) in the legend

*Line 253: Tilting.*

- We have corrected this here, and in a couple of other places in the text

*Line 268-269: Do not use too many perhaps. One of the best examples is.....*

- We have rephrased the start of the sentence as suggested, it now reads as follows: "One of the best examples may be the onset of microbial carbonate deposition in hypersaline basins …"

*Line 284: Mw 7.*

- It is corrected

*Line 289: Delete "significantly"*

- Not sure why it needs to be removed, it seems to be a useful addition to the sentence:
- "Fault networks in rifts are also less well ordered, because they accumulate **significantly** less strain when compared to subduction zones."

*Line 316: Shanxi rifts. Weihe rift is part of it.*

- Thanks for noticing, we now only refer to the Shanxi rift as a whole, and modified the subsequent sentence to reflect this change.

*Line 350: is->are*

- It is corrected

*Line 378: Delete just.*

- We decided to remove the reference to Eritrea altogether, as we describe the Afar Rift as a region here, which is split between Eritrea, Ethiopia, and Djibouti. As such, we would have to mention all countries and this will become superfluous, we felt. The text now reads as follows:
  - o "… the 2011 eruption of the Nabro volcano caused several fatalities (Goitom et al., 2015). Moreover, the 2011 Nabro eruption resulted in the expulsion of 1.5 megatons of $SO_2$, ranking it as the largest eruption since that of Mount Pinatubo in 1991 …"

*Line 387: Delete especially.*

- It is deleted here. NB: We also reworded "especially" in (old) Line 851, which now reads: "This is of particular relevance when it comes to intra-plate earthquakes …"

*Line 401: Delete occurrence of.*

- It is deleted

*Line 471: Delete "Still, perhaps"*

- We have replaced it with "However", as there needs to be some indication of contradiction at the start of this sentence.

*Line 474: Delete such.*

- It is deleted

*Line 492: extensional*

- Thanks for noticing, it is corrected

*Line 498: Delete the comma after environments*

- It is deleted

*Line 522: What do you mean "base metals important to energy transition"?*

- From Blowes et al. (2014), which is added as a reference:
  - "Base metal is a wide-ranging term that refers either to metals inferior in value to those of gold and silver or, alternatively, to metals that are more chemically active than gold, silver, and the platinum metals (AGI, 1957). Accordingly, a review of base-metal mineralogy would encompass much of the world's metal production and geology. Usage of the 'base metal' term in the mining and minerals industry is rather loose, but a common application is to the nonferrous metals excluding precious metals. These include copper, lead, zinc, nickel, and tin. Kesler (1994), however, grouped nickel with ferroalloy metals along with manganese, chromium, silicon, cobalt, molybdenum, vanadium, tungsten, niobium, and tellurium and copper, lead, zinc, and tin as base metals. Among the base metals, tin is by far the least significant in terms of volumes consumed and monetary value."
- The sentence now runs as follows:
  - "Sediment-hosted mineral deposits are the largest global resource of lead and zinc (Goodfellow et al., 1993; Mudd et al., 2017), both base metals important to the global transition to renewable energy (Blowes et al. 2014; IEA, 2021).

*Line 526: What is a focussing pathway?*

- The idea is that faults can focus upward fluid flow, we have rephrased the sentence to avoid confusion:

o "… which could have provided permeable pathways focussing upward fluid flow …"

*Line 535-537: This sentence is too loose to read. Rephrase it.*

- We have rephrased things, while splitting the sentence in two parts. It now runs as follows:
    o "By finding correlations between geophysical and geological observational datasets and the locations of known CD-type and MVT deposits, researchers try to limit the exploration space for these deposits (e.g., Hoggard et al., 2020; Lawley et al., 2022; Burisch et al., 2022). For example, Hoggard et al. (2020) found that all giant deposits lie within 200 km of the edges of cratonic lithosphere."

*Line 544: Delete happening.*

- It is deleted

*Line 553: What does "such" stand for? Be more specific.*

- We have deleted "such" in the sentence, as it was the first discovery of hydrothermal venting on the sea floor, thus avoiding confusion. Moreover, we have slightly modified the end of the sentence, so that it now reads as follows:
    o "The first seafloor hydrothermal activity was discovered in the Red Sea oceanic spreading centre (Miller et al. 1966), and many more sites have been identified afterwards"

*Line 558: "or the other way around". Not clear. Delete or rephrase.*

- We have deleted it to avoid too long a sentence.

*Line 562: Delete magmatic, and above old and thick. Repeated.*

- It is rephrased as follows:
    o "Kimberlites are rocks formed from highly volatile ultramafic eruptions in cratons, shields and mobile belts …"

*Line 566: Delete "break-up, specially"*

- It is deleted

*Line 568: Delete those of, and which was emplaced*

- The sentence is rephrased to:
    o "Other magmatic ore deposits linked to rifting are layered intrusions such as those of the Bushveld Complex in the Kaapvaal Craton, which likely formed in a back-arc setting and contain, amongst others, copper, nickel and platinum-group metal deposits (Clarcke et al. 2009)(Fig. 11)."

*Line 683: Icelandic->Iceland*

- We believe that "Icelandic", as "in the Icelandic case" is fine here

*Line 684: Too oral for set up camp.*

- We now use "have established themselves"

*Line 769: How to define "fresh"? Altered rocks may have undergone carbonation.*

- As "unaltered", we have now defined this as follows:
  - "… $CO_2$ can also be chemically stored when it reacts with unaltered (fresh) exhumed rocks, a process known as carbonation …"

*Line 773: Mantle rocks are ultramafic, not mafic.*

- We now write "(ultra)mafic" to reflect this nuance

*Line 777: Delete the first and second "the"*

- They are deleted

*Line 786: Now-extinct means abandoned or ancient? Be clear.*

- We meant "abandoned" and have corrected the text accordingly